# Circadian clock features define novel subtypes among breast cancer cells and shape drug sensitivity

Carolin Ector [1,2,3], Jeff Didier [4], Sébastien De Landtsheer[4], Malthe S Nordentoft[5], Christoph Schmal[6], Ulrich Keilholz[1,7], Hanspeter Herzel [6,8], Achim Kramer [9], Thomas Sauter [4] & Adrián E Granada [1,7 ✉]

## Abstract

The circadian clock regulates key physiological processes, including cellular responses to DNA damage. Circadian-based therapeutic strategies optimize treatment timing to enhance drug efficacy and minimize side effects, offering potential for precision cancer treatment. However, applying these strategies in cancer remains limited due to a lack of understanding of the clock's function across cancer types and incomplete insights into how the circadian clock affects drug responses. To address this, we conducted deep circadian phenotyping across a panel of breast cancer cell lines. Observing diverse circadian dynamics, we characterized metrics to assess circadian rhythm strength and stability in vitro. This led to the identification of four distinct circadian-based phenotypes among 14 breast cancer cell models: functional, weak, unstable, and dysfunctional clocks. Furthermore, we demonstrate that the circadian clock plays a critical role in shaping pharmacological responses to various anti-cancer drugs and we identify circadian features descriptive of drug sensitivity. Collectively, our findings establish a foundation for implementing circadian-based treatment strategies in breast cancer, leveraging clock phenotypes and drug sensitivity patterns to optimize therapeutic outcomes.

**Keywords** Circadian Clock; Circadian Medicine; Systems Biology; Breast Cancer
**Subject Category** Cancer

## Introduction

In alignment with the solar 24-h day–night cycle, the circadian clock regulates essential physiological and behavioral processes in almost all organisms. In mammals, a hierarchical structure coordinates rhythmic activities across both whole-organism and cellular levels (Chaix et al, 2016; Golombek et al, 2014), with the master clock residing in the suprachiasmatic nucleus (SCN) and

individual peripheral clocks oscillating across tissue types (Chaix et al, 2016). Remarkably, about 40% of all genes are rhythmically expressed in a tissue-specific manner (Zhang et al, 2014), influencing various biological functions such as metabolism (Neufeld-Cohen et al, 2016), cell growth (Chakrabarti and Michor, 2020), immune responses (Scheiermann et al, 2013), and DNA repair (Sancar et al, 2010), thereby maintaining cellular balance.

At its core, the circadian mechanism is composed of transcriptional-translational feedback loops (TTFLs) (Takahashi, 2017). These TFFLs consist of the CLOCK and BMAL1 transcription factors, which induce the transcription of *Per1/2/3* and *Cry1/2* genes through the binding to the respective promoter regions. In turn, PER and CRY proteins inhibit the binding of CLOCK/BMAL1, and consequently their own transcription. In a complementary second feedback loop the transcription of *Bmal1* is rhythmically regulated by REV-ERBα/β repressor and RORα/β/γ activator proteins, whose transcription is likewise dependent on the CLOCK/BMAL1 transactivation complex (Chaix et al, 2016; Takahashi, 2017).

Environmental factors are crucial in maintaining the synchronization of circadian rhythms in our body. Misalignments of this synchronization, for example through prolonged exposure to shift work, are associated with various diseases, including cancer (Sulli et al, 2019)—the second-leading cause of death worldwide (https://www.who.int/health-topics/cancer, 2022). In-depth studies have further shown how the disturbance of the circadian clock, through either genetic mutations or the suppression of core clock genes is directly linked to the progression of cancer and poorer survival rates (Ye et al, 2018). The clock's influence extends to cancer therapy, affecting the efficacy and toxicity of treatments in a time-of-day-dependent manner (Lee et al, 2021; Ye et al, 2018). Despite its high medical importance, the distinct role of the circadian clock in cancer remains poorly understood, highlighting the need for further research to optimize treatment strategies by circadian rhythms.

To illuminate the circadian clock's role in cancer, our study focuses on breast cancer, the most prevalent cancer among women and the second most common overall (Bray et al, 2024). Breast cancer is categorized into four main subtypes, distinguished by the cell phenotype (basal or luminal) and specific biomarkers,

[1]Charité Comprehensive Cancer Center, Charité – Universitätsmedizin Berlin, 10117 Berlin, Germany. [2]Berlin School of Integrative Oncology, Charité – Universitätsmedizin Berlin, 10117 Berlin, Germany. [3]Faculty of Life Sciences, Humboldt-Universität zu Berlin, 10115 Berlin, Germany. [4]Department of Life Sciences and Medicine, University of Luxembourg, L-4365 Esch-sur-Alzette, Luxembourg. [5]Niels Bohr Institute, University of Copenhagen, 2100 Copenhagen, Denmark. [6]Institute for Theoretical Biology, Humboldt-Universität zu Berlin, 10115 Berlin, Germany. [7]German Cancer Consortium (DKTK), Berlin, Germany. [8]Charité – Universitätsmedizin Berlin, 10117 Berlin, Germany. [9]Laboratory of Chronobiology, Charité – Universitätsmedizin Berlin, 10117 Berlin, Germany. ✉E-mail: adrian.granada@charite.de

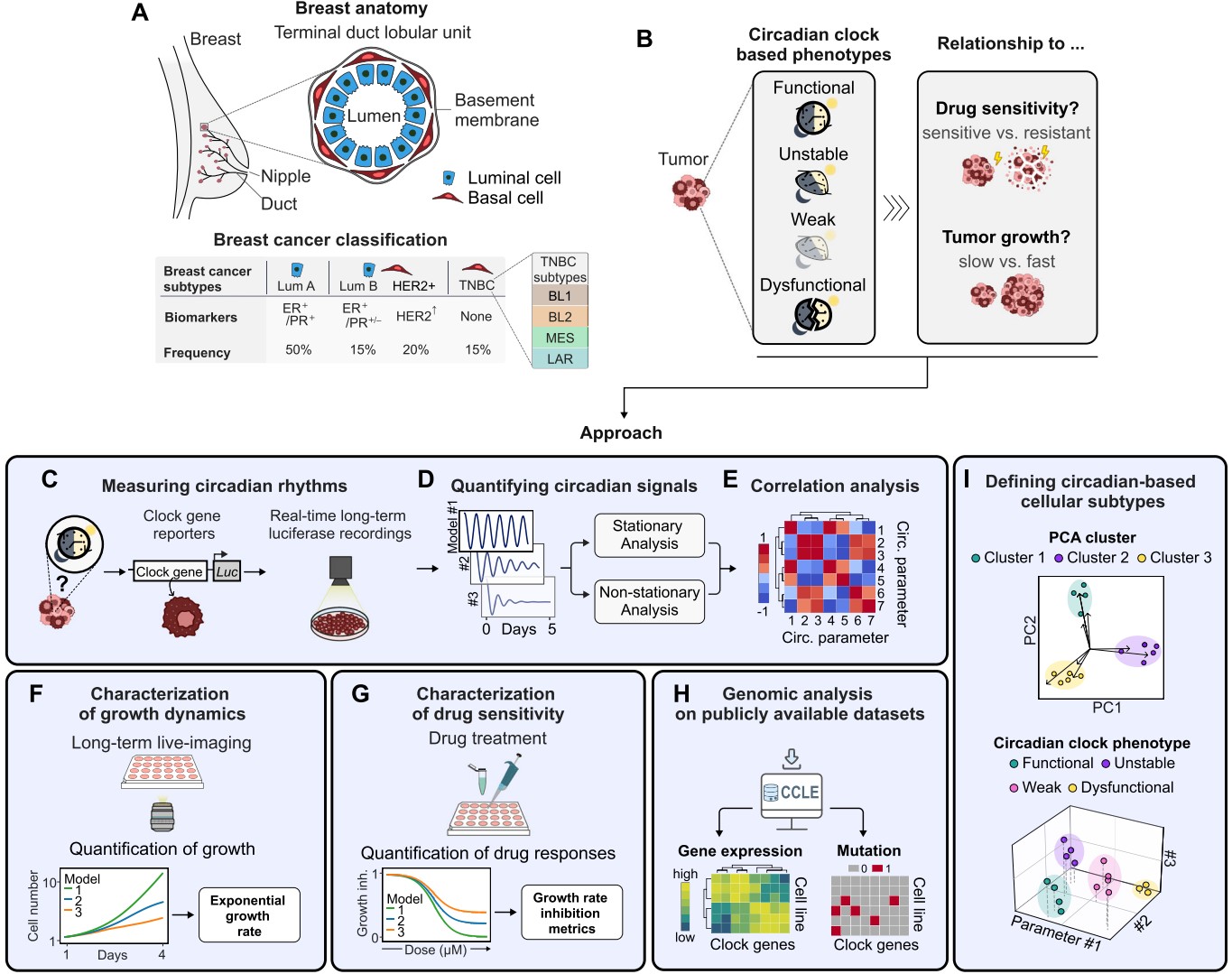

**Figure 1.  Definition of circadian-based breast cancer subtypes by deep phenotyping.**

(A) Classification of breast cancer based on cancer cell phenotype and biomarker expression status. Approximate frequency among all breast cancer cases indicated in percentage. −, negative; +, positive; ER, estrogen receptor; PR, progesterone receptor; HER, human epidermal growth factor receptor 2; LumA/B, Luminal A/B; TNBC, triple-negative breast cancer, BL1/2, basal-like 1/2; MES, mesenchymal-like; LAR, luminal androgen receptor. (B) Aim of our study to identify circadian-based phenotypes in breast cancer cell models and investigate the role of circadian clock features for tumor growth and drug sensitivity. (C) Cellular circadian rhythms are measured by long-term luciferase (Luc) recordings of clock gene reporter cell lines. (D) Circadian signals are quantified by stationary and non-stationary analysis approaches. (E) Circadian parameters are related to each other by correlation analysis to define a set of representative circadian metrics. (F) Long-term live imaging is employed to capture growth dynamics across various cell line models. (G) Characterization of drug sensitivities across various cell line models, followed by the parametrization of growth rate inhibition metrics. (H) Characterization of genomic circadian profiles of the cell models tested, utilizing publicly available datasets. (I) Definition of circadian-based cellular subtypes through the integration of multiple cellular parameters.

including progesterone and estrogen receptors, as well as HER2-receptor overexpression (Sotiriou et al, 2003) (Fig. 1A). Within the highly heterogeneous and aggressive triple-negative breast cancer (TNBC) subtype lacking all three biomarkers, Lehmann et al, identified four molecular TNBC subtypes: basal-like 1 (BL1) and 2 (BL2), mesenchymal-like (MES) and immunomodulatory (IM) and luminal androgen receptor positive (LAR) (Lehmann et al, 2011; Lehmann et al 2016). While the presence of circadian rhythms have been shown for selected breast cancer models (Lellupitiyage Don et al, 2019; Li et al, 2024), the general assumption is that more aggressive breast cancer subtypes including TNBC exhibit disrupted circadian rhythms (Lellupitiyage Don et al, 2019; Li et al, 2024; Rida et al, 2019). In this study, we aim to define novel circadian clock-based phenotypes within breast cancer and to uncover how circadian properties relate to drug sensitivity and tumor growth (Fig. 1B). For this, we employ a comprehensive set of data analysis methods to deep-phenotype the circadian clock (Fig. 1C–E), and to collect information on cellular growth dynamics (Fig. 1F), drug responses (Fig. 1G), and genomic properties of circadian clock genes (Fig. 1H). Subsequently, we integrate our acquired cellular parameters to identify cell clusters and establish novel circadian-based subtypes within breast cancer (Fig. 1I).

Finally, we investigate how drug responses depend on the circadian clock, identifying several drugs that could be further explored for their potential in chronotherapeutic approaches. Altogether, our approach sets the stage for a deeper understanding of distinct tumor-specific circadian clocks and how they shape cancer therapy outcomes.

# Results

## Deep circadian phenotyping reveals variability in clock strength across breast cancer models

The role of the circadian clock in cancer progression and treatment is gaining increasing interest, yet the degree of circadian rhythmicity and intrinsic timing profiles for various cancer subtypes remains largely unclear. To address this, we utilized high-resolution circadian clock recordings alongside a range of time-series analysis techniques to effectively measure and characterize circadian rhythms in cancer models. Building on our previously described deep-circadian phenotyping approach (Ector et al, 2024), which is comprehensively detailed in Box 1, we expanded our dataset by incorporating additional cell line models and novel analysis methods.

To obtain an initial estimate of the periodic quality of the time-series, we commenced our deep circadian phenotyping approach with autocorrelation (AC) analysis. Screening of 19 different cancer and non-cancer cell lines with the majority classifying as the highly heterogeneous and aggressive TNBC, we identified a broad spectrum of rhythmicity indices and period lengths, mostly centered around the 24-h human circadian cycle. Interestingly, rhythmicity indices below 0.3, indicative for weaker rhythms, often deviated from 24 h, whereas higher indices aligned with the circadian cycle (Fig. 2A). These results indicate a considerable variability of circadian phenotypes within the cell lines tested, highlighting the need to further dissect the circadian characteristics.

To quantify the proportion of distinct frequency components, present in each signal, we employed multiresolution analysis (MRA). This process segmented the detrended signal into four frequency bands: noise (1–4 h), ultradian (4–16 h), circadian (16–32 h), and infradian (32–256 h) (see Methods). Relating the circadian to the noise component revealed a spectrum of signal-to-noise ratios across cell lines, where higher circadian signals corresponded to lower noise, indicating circadian signal robustness, and vice versa. Alongside the osteosarcoma cell line U-2 OS, which is well-studied for its functional circadian clock (Baggs et al, 2009), high signal-to-noise ratios were found for the epithelial MCF10A, the LumA-MCF7 and the TNBC-MDAMB468 cell line (Fig. 2B). Interestingly, a knockout of *Cry2* in the U-2 OS cell line alone did not essentially alter the signal-to-noise ratio. In contrast, the knockout of *Cry1* or both, *Cry1* and *Cry2*, decreased the circadian component substantially by 34.9% ($p = 5.3 \times 10^{-6}$) and 59.8% ($p = 8.9 \times 10^{-8}$), while significantly enhancing the noise component by 3.3-fold ($p = 9.0 \times 10^{-5}$) and 120.1-fold ($p = 1.2 \times 10^{-5}$), respectively.

## Clock dynamics vary within subtypes of breast cancer

After evaluating the strength and composition of rhythmic structures within our signals, we analyzed their dynamic nature through continuous wavelet transform (CWT). This approach visualizes the signal in a CWT power spectrum heatmap, showcasing the range and relative power of signal components over time within a specified period range. By tracking the main oscillatory component ("*ridge*"), CWT effectively reveals non-stationary and time-dependent features, such as dynamic periods and amplitudes (see Appendix Fig. S1A and Methods).

Aggregating the distribution of weighted mean periods and amplitudes per subtype, we noted variability of these parameters over time, particularly for the three TNBC subtypes (Fig. 2C; see Appendix Fig. S1B and Methods for weighting approach). Furthermore, we revealed varying prevailing periods and amplitudes for each subtype where only the osteosarcoma subtype showed a distinct period peak at approximately 24 h, combined with the highest amplitudes, while the prevailing periods for the other subtypes were mainly prolonged (Fig. 2C). Focusing on the breast cancer subtypes, a clear differentiation in circadian properties was visible, where the more aggressive TNBC subtypes exhibited longer periods and lower amplitudes than the LumA subtype (Fig. 2C). Though, when considering median CWT periods and amplitudes of each cell model, we found considerable within-subtype diversity, indicating that individual cancer tissue types exhibit a spectrum of circadian clock phenotypes (Appendix Fig. S1C). This within-subtype diversity extends to the ridge lengths, a measure of circadian clock strength, where longer and continuous ridges denote robust signals and shorter, discontinuous ridges indicate weaker ones (Fig. 2D). Here, only TNBC-M cell models ranked alongside of each other, whereas cell models of the LumA and basal-like TNBC subtypes displayed considerable variability in ridge lengths.

Building on the identified temporal circadian dynamics within and across cancer subtypes, we next assessed *Bmal1-Per2* phase differences over time. Consistent with the previously discussed circadian parameters, we observed a spectrum of phase differences across and within tissue types (Fig. 2E,F). Despite this variability, a notable pattern emerged in which non-cancerous and osteosarcoma tissues consistently showed phase differences around $2\pi/3$, an 8-h lag in a 24-h cycle, in contrast to breast cancer subtypes, which tended to have no phase difference ($2\pi$) (Fig. 2F).

Circadian clocks in breast cancer cell lines may exhibit either self-sustained or damped oscillatory behaviors, a distinction that depends on the degree of intercellular coupling. Aiming to infer intercellular circadian coupling strength, we next analyzed signal amplitudes and corresponding exponential decay rates. Inspired by previous studies (Del Olmo et al, 2024; Finger et al, 2021; Guenthner et al, 2014), we used a network of identical Poincaré oscillators to model individual cells within a tissue, featuring a constant coupling strength (κ) and periods averaging a 24-h cycle. By varying the coupling strength, we observed distinct signal patterns, with strong coupling leading to amplified and self-sustained oscillations, while weak coupling results in a damped oscillation pattern (Fig. EV1).

We then fitted our experimental data to an exponentially decaying sinusoidal function (Fig. 2G and Methods) to determine initial amplitudes ($A_0$) and amplitude decay rates ($\gamma$). A comparative analysis unveiled an L-shaped trend in both, the simulated and experimental datasets, facilitating the approximation of coupling strengths in experimental data (Fig. 2H). Here, stronger intercellular coupling was identified for signals with high

**Box 1  Methods and techniques for deep circadian phenotyping**

**(A)** Schematic of the experimental and computational approach to deep phenotype circadian rhythms. Stable circadian reporter cell lines expressing Luciferase reporters for either *Bmal1* or *Per2*, two core clock genes of the circadian clock network, were generated by lentiviral transduction. The circadian clocks of individual cells in a cell population were reset, and signals from the reporters were monitored in real-time by bioluminescence recordings spanning multiple days.

**(B)** Raw signal (left plot) and pre-processed signal traces (middle and right plots) of the TNBC cell line HCC1143 *Bmal1*- and *Per2*-Luc. Pre-processing of raw signals involved detrending (middle plot) and, for parts of the analysis, amplitude-normalization (left plot).

**(C)** Autocorrelation analysis of detrended HCC1143 *Bmal1*- and *Per2*-Luc signals. The arrow indicates the rhythmicity index and corresponding period of the time-series (lag). Dashed lines = 95.4% CI.

**(D)** Multiresolution analysis of detrended HCC1143 *Per2*-signal to decompose the signal into four frequency components, % = fraction of each component to the total signal.

**(E)** Continuous wavelet transform analysis on detrended and amplitude-normalized HCC1143 *Per2*-signal (top left plot), showing time-resolved signal periods in a wavelet spectrum (bottom left plot). The red line marks the main oscillatory component (ridge). The right panel shows the corresponding ridge readout for the period (top), amplitude (middle), and phase difference (difference to HCC1143 *Bmal1*-signal (bottom).

**(F)** Approach to calculate the coupling strength (κ) of circadian oscillators within a population of cells from the signal's amplitude ($A_o$) and exponential decay rate (γ).

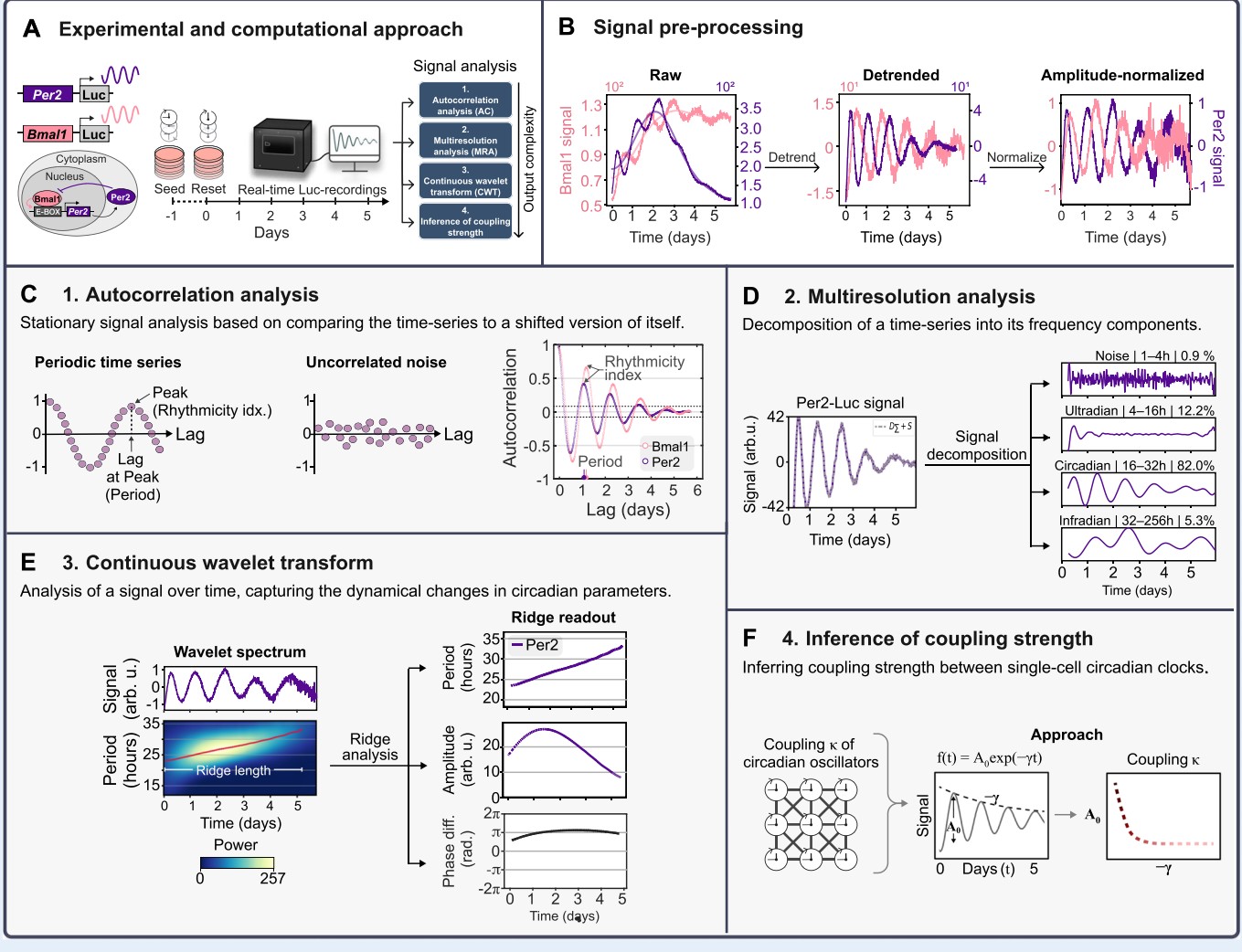

amplitudes and slow decay rates and vice versa. These findings align with previous studies, which demonstrated that a reduction in coupling strength results in increased variance, disrupting both the summed amplitude and period (Del Olmo et al, 2024; Guenthner et al, 2014). Furthermore, focusing on U-2 OS cells, we observed a

significant decrease in coupling strength in the circadian knockout cell lines compared to the wild-type (Fig. 2H), which is consistent with predictions from earlier research (Del Olmo et al, 2024).

These insights into the dynamics of circadian clock features, together with the broad range of circadian rhythm strengths

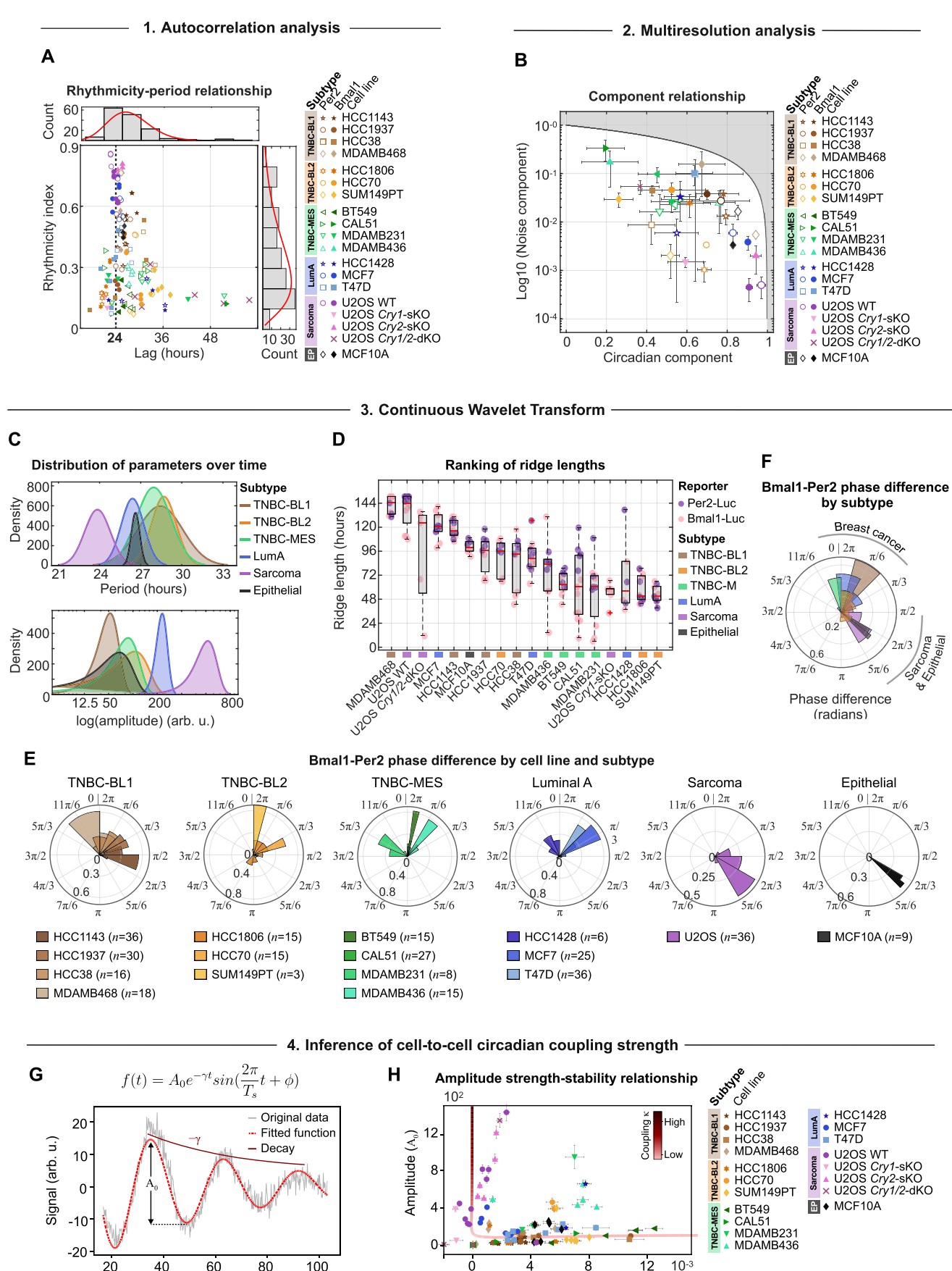

**Figure 2. Deep phenotyping of circadian rhythms in breast cancer cell line models.**

(A) Relationship between rhythmicity indices and periods for various cell line models, distinguished by markers and colored by subtype ($n = 141$ samples represented as individual measurements). Dashed line at period = 24 h. Gamma distribution of periods and rhythmicity indices shown in histograms in the top and left panel, respectively. (B) Relationship between normalized noise and circadian components across cell line models, determined by multiresolution analysis. Cell models are color-coded by subtype and distinguished by markers. Shaded area covers unattainable range. Data represents the mean ± s.d. of 6 samples per reporter ($n = 2$ biological replicates á technical triplicates or duplicates [HCC1806 Per2-Luc, $n = 5$]). $n = 3$ for MCF10A and MDAMB468-Per2-Luc (single experiment). (C) Normal distribution of subtype-averaged and time-resolved periods (top panel) and amplitudes (bottom panel) ($n = 1-4$ cell lines models per subtype, see legends of (A) and (B) for exact numbers). Before calculating subtype averages, all available replicates per cell model were averaged ($n = 3-6$, see (B) for exact numbers) and used as input. (D) Boxplot of CWT ridge lengths from Bmal1- and Per2-signals of specified cell models, with the bottom and top edges of the boxes representing the 25th and 75th percentiles, respectively. Extending whiskers represent data points within 1.5 times the interquartile range from lower and upper quartile. Median values are marked by red horizontal lines, and outliers by red crosses. $n = 4-12$ samples per cell line (see (B) for exact numbers). (E) Subtype-specific polar histograms of Bmal1-Per2 phase differences over time, averaged across multiple replicates and color-coded by cell line. The number $n$ of Bmal1-Per2-combinations per cell line is indicated next to the cell line name. Polar histograms were normalized by probability. (F) Polar histogram of Bmal1-Per2 phase differences over time, averaged and colored by subtype ($n = 1-4$ cell line models, see (E) for exact numbers). In the polar histograms in (E) and (F) $2\pi$ corresponds to one full circadian cycle. (G) Example of fitting signals to an exponentially decaying sinusoidal model with constant periods to deduce initial amplitudes ($A_O$) and amplitude decay rates ($\gamma$). Dark red amplitude decay trace is sketched for illustration purposes. (H) Relationship between initial signal amplitudes and amplitude decay rates across samples of different cell line models ($n = 117$ samples). Cell models are color-coded by subtype and distinguished by markers. Error bars represent the standard deviation of the fitted parameters, reflecting their uncertainty as determined by the fitting process. The L-shaped line represents the simulated coupling strength ($\kappa$). Source data are available online for this figure.

previously described, underscore the heterogeneity of circadian regulation within cancer tissues. Such within subtype-diversity motivated us to establish a novel circadian-based classification that could potentially inform chronotherapeutic strategies.

## Establishing a novel circadian-based subtyping in breast cancer

After extracting various circadian clock parameters from the different cell lines, we next investigated whether circadian clock-based clusters are present among these lines and how this clustering relates to their classical subtypes. Considering the established molecular connections between the circadian clock and cell cycle (Feillet et al, 2015; Gonze, 2024; Gutu et al, 2024), we integrated cellular growth rates into our analysis, which we determined by exponential curve fitting of long-term growth data (Appendix Fig. S2). To streamline our extensive dataset, we initially assessed the relationships between the parameters by calculating pairwise Pearson correlation coefficients, aggregating data from all replicates per cell line and from both circadian reporters. In line with our expectations, we discovered several strong correlations as well as redundancies across the dataset (Fig. EV2), allowing us to narrow down the circadian dataset to key parameters shown in Fig. 3A. We observed significant positive correlations between the different clock strength metrics, highlighting consistency across the different signal analysis approaches. In contrast, clock-strength parameters were mostly negatively, yet weaker, correlated with the growth rate, demonstrating accelerated growth in cell models with weaker circadian rhythms, whereas parameters representative of clock instability such as period and phase difference variations over time were positively correlated with growth (Fig. 3A and Fig. EV2A).

Upon refinement of our dataset, we employed principal component analysis (PCA) to investigate the presence of circadian-based clusters within the breast cancer and epithelial cell line panel. PCA showed the formation of three distinct circadian-based clusters where the respective cell models share characteristics of circadian clock or growth dynamics (Fig. 3B). Notably, most of the variability in the dataset was captured by the first two principal components alone, which accounted for 54.7% and 21.1% of the

variance, respectively. Ranking of parameters by their corresponding PC loading, revealed that the highest influence on the observed clustering for the first principal component comes from the infradian and circadian components, underlining their potential as key markers in the circadian characterization of the cell models (Fig. EV2B). Interestingly, the circadian-based grouping deviated from the classical subtyping, providing a novel perspective on the subtyping of these cell models.

Since the identified PCA clusters do not directly translate into circadian phenotypes, we selected one key circadian parameter of each of the identified PCA clusters and performed k-means cluster analysis (Fig. 3C). In detail, we focused on the oscillation period, the Bmal1-Per2 phase difference variability over time for clock stability, and the circadian component as a measure of clock strength. K-means clustering classified the cell models into four circadian phenotypes based on their circadian clock functionality: functional, unstable, weak, or dysfunctional (Fig. 3C). The functional phenotype, marked by a high circadian component and stable phase difference, suggests a strong and consistent circadian rhythm, and is composed of the epithelial MCF10A, two LumA, and two TNBC-BL1 cell line models. In contrast, the dysfunctional phenotype, comprising of two cell models only, is characterized by instable phase differences over time, long periods and low circadian bands. The intermediate phenotypes, weak and unstable, display decreased rhythm strength and stability, respectively (Fig. 3C). The decision to use four clusters was guided by the elbow method, analyzing the relationship between the within-cluster sum of squares (WCSS) and the number of clusters (k). The point of diminishing returns was observed at k = 4, suggesting it as the optimal number of clusters, which was further reflected by the greatest silhouette score of 0.445 (Appendix Fig. S3). Consistent with k-means clustering, cell models sharing similar circadian clock phenotypes were mapped in proximity using Uniform Manifold Approximation and Projection (UMAP) with varying dimensions ($n = 2$, 3, and 4) (Fig. EV2C).

In summary, we introduced a new phenotypic framework that categorizes breast cancer cell lines according to their circadian phenotypes. This approach provides a complement to traditional subtyping methods, potentially enhancing the evaluation of cancer models for circadian-based therapeutic strategies.

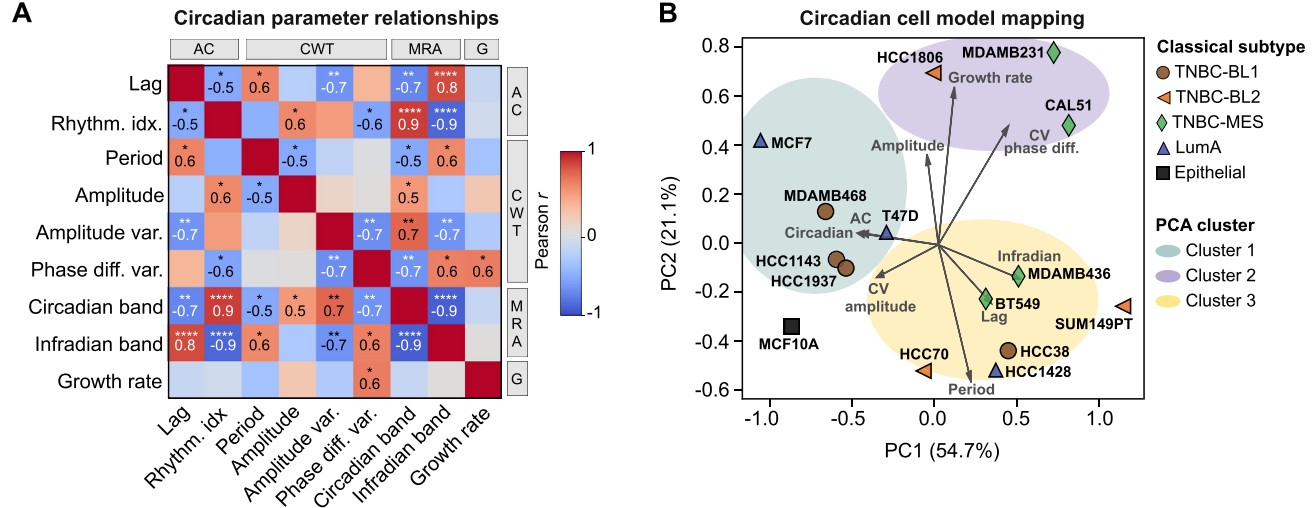

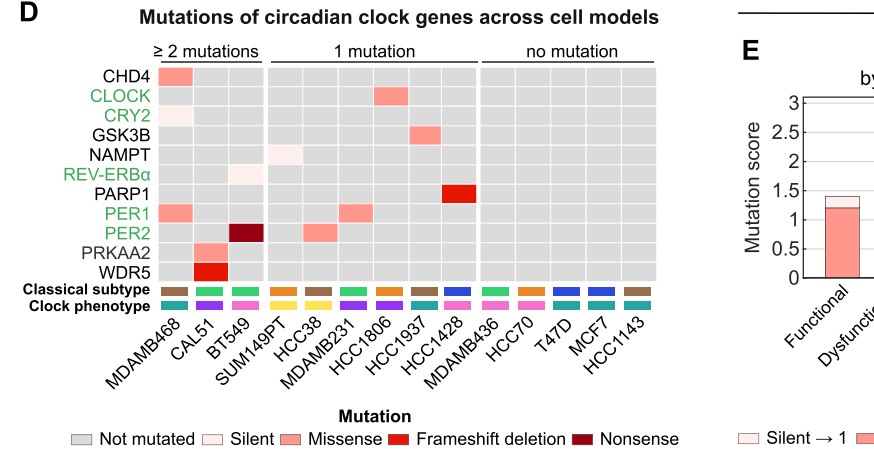

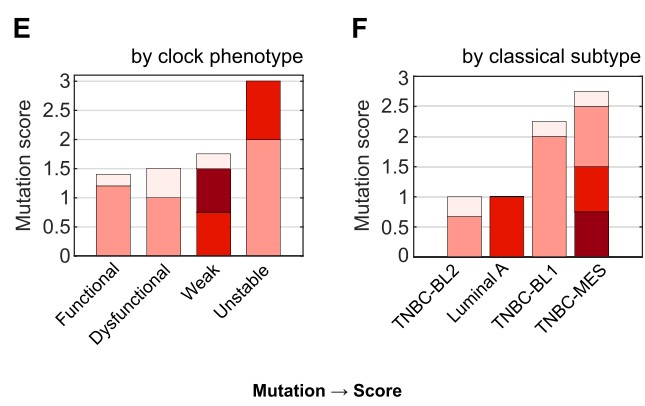

## Genetic profiles of circadian-based phenotypes

To evaluate the role of circadian genetic profiles in the newly identified circadian-based phenotypes, we extracted the mutation and gene expression profiles of 16 core clock genes and 44 circadian clock-related genes from the cancer cell line encyclopedia (Barretina et al, 2012) (Dataset EV1). We specified core clock genes as those essential for regulating the circadian TTFLs and for

**Figure 3.  Circadian clock features define circadian-based subtypes in breast cancer cell line models.**

(A) Pearson correlation coefficients between selected *Bmal1-Per2*-averaged circadian features and growth rates across all breast cancer cell line models and the epithelial MCF10A cell line ($n = 15$ cell lines). Parameters are categorized by their approach of calculation: AC, autocorrelation; CWT, continuous wavelet transform; MRA, multiresolution analysis; G = growth. Displayed are statistically significant correlation values, where *, **, ***, and **** indicate *p*-values < 0.05, 0.01, 0.001, and 0.0001, respectively. Exact *p*-values: Rhythm. idx.-Lag and Amplitude-Period = $3.2 \times 10^{-2}$; Period-Lag = $1.4 \times 10^{-2}$; Amplitude-Rhythm. idx. and Phase diff. var.-Rhythm. idx. = $2.1 \times 10^{-2}$; Amplitude var.-Lag = $6 \times 10^{-3}$; Circadian band-Lag and Circadian band-Amplitude var. = $2 \times 10^{-3}$; Circadian band-Rhythm. idx. And Infradian band-Rhythm. idx. = $1.1 \times 10^{-5}$; Circadian band-Period = $2.9 \times 10^{-2}$; Circadian band-Amplitude = $4.6 \times 10^{-2}$; Circadian band-Phase diff. var. and Infradian band-Amplitude = $6 \times 10^{-3}$; Infradian band-Lag = $1 \times 10^{-4}$; Infradian band-Period = $2 \times 10^{-2}$; Infradian band-Phase diff. var. and Growth rate-Phase diff. var. = $1.2 \times 10^{-2}$; Infradian band-Circadian band = $1 \times 10^{-6}$. (B) Principal component analysis (PCA) biplot of the indicated circadian and growth parameters, displaying the distribution of 15 cell line models across the first two principal component scores. Cell models are depicted by different markers and colors indicating their classical subtype. Color-coded clusters are outlined around closely positioned cell models, based on visual inspection. The variance (PC loadings) explained by each component is expressed in percentage. CV = coefficient of variation. (C) Three-dimensional k-means clustering ($k = 4$) of the indicated cell line models. Discretization was based on three min–max-scaled circadian parameters from each PCA-cluster defined in subplot (B). Novel clock phenotypes (distinct clusters) are color-coded. Classical subtypes are denoted with different markers (refer to subplot (B) for key). Within cluster sum of squares = 0.79, cluster silhouette score = 0.445. CV, coefficient of variation. (D) Cell line-specific mutation map, color-coded by type, of core circadian clock (green) and circadian clock associated genes (black). Color-coded rectangles above the cell line names indicate the classical subtype and clock phenotype (refer to subplot (B) and (C) for color-coding). See Dataset EV1 for full list of investigated circadian genes. (E) Ranking of mutation scores, normalized by cell line model per clock phenotype. (F) Ranking of mutation scores, normalized by cell line model per classical subtype. In (E) and (F), stacked bars show the proportion of each mutation type contributing to the final score, $n = 2$–4 cell models per category; for exact numbers of cell line models per group refer to (C). Source data are available online for this figure.

which any dysregulation or mutation can lead to disrupted circadian rhythms (Ko and Takahashi, 2006; Takahashi, 2017).

We noted varying mutation profiles of the circadian genes across the 14 tested breast cancer cell lines, with the majority of cell models exhibiting at least one mutation, whereas five cell lines showed no mutations in their circadian clock network (Fig. 3D). We then categorized mutations based on their impact on the protein sequence, assigning scores from 1 for silent mutations to 3 for highly damaging mutations like nonsense and frameshift mutations. Thereby we could compare the clock phenotypes by their circadian mutation burden, revealing that the unstable clock phenotype exhibits the highest mutation score while the functional phenotype had the lowest one (Fig. 3E). Notably, damaging mutations appeared exclusively in the unstable and weak clock phenotypes, suggesting a dysregulation of the circadian clock mechanism. Contrary to our expectations, the dysfunctional clock phenotype shared a similar low mutation burden with the functional phenotype, indicating that the number and severity of mutations alone may not fully explain clock functionality, and that additional factors might play a role for defining the underlying clock phenotypes (Fig. 3E). When analyzing the mutation scores across classical cancer subtypes, we found that the TNBC-mesenchymal subtype had the highest mutation burden, while TNBC basal-like 2 had the lowest one (Fig. 3F). We also identified a tendency for less damaging mutations in basal-like subtypes, whereas Luminal A and TNBC-mesenchymal subtypes exhibited more damaging mutations in the circadian gene panel.

We next sought to identify whether expression patterns of core circadian clock genes align with the identified clock phenotypes. To do so, we clustered the cell models based on their circadian gene expression dynamics and exposed three overarching clusters (Fig. EV2D). However, these clusters did not reflect the clock phenotypes but rather showed a distinction between the LumA and the TNBC subtypes. This distinction also emerged when mapping the cell models in a dimensionality-reduced UMAP-space (Fig. EV2E), indicating that the difference in circadian gene expression dynamics is associated with underlying breast cancer subtypes rather than the intrinsic circadian phenotype. It is important to note that overall transcriptional activities vary significantly between breast cancer subtypes (Perou et al, 2000; Rida et al, 2019). These differences in general transcriptional activity may influence the expression levels of

core clock genes independently of their circadian functionality, reinforcing our observed gene expression variations reflecting subtype-specific transcriptional programs rather circadian phenotypes.

## Circadian features drive sensitivity profiles to numerous drugs

Due to several interactions between circadian clock genes and therapeutic drug targets, circadian clock-aligned treatments have proved to increase drug effects based on the optimal time-of-day at their administration (Dallmann et al, 2014; Lee et al, 2021; Ye et al, 2018). To unravel the crucial link between circadian features and drug sensitivity, we employed linear regression and Pearson correlation analysis on a set of sensitivity parameters to various drugs and circadian oscillation features. To account for cell division events during drug perturbation assays and to obtain robust sensitivity metrics, we assessed drug sensitivity using the normalized growth rate inhibition (GR) approach (Hafner et al, 2016). For this, we expanded the analysis of our drug-response dataset published in Ector et al, (Ector et al, 2024). We analyzed a broad spectrum of drugs including DNA synthesis inhibitors (5-FU, doxorubicin), mitosis inhibitors (alisertib, paclitaxel), inhibitors for PI3K (alpelisib), or mTOR (torin2), and DNA damage response (DDR) targeting agents (adavosertib, cisplatin, olaparib). Through pairwise linear correlation analysis, we discovered a significant anti-correlation between the maximal effect of the DDR-inducer cisplatin and the autocorrelation rhythmicity index ($r = -0.64$, $R^2 = 0.41$) (Fig. 4A). Here, higher rhythmicity was associated with more cytotoxic responses to cisplatin, whereas lower rhythmicity was linked to cytostatic or less toxic responses. This anti-correlation extended to other circadian strength-related parameters such as the amplitude, ridge length and circadian component (Fig. EV3A). Aligning with these observations, weaker responses to cisplatin were positively correlated with clock instability parameters, such as the variability in periods and phase differences. Across all drugs tested, we identified additional significant relationships between circadian metrics and $GR_{inf}$ values. Notably, $GR_{inf}$ values from alpelisib showed the strongest correlation with circadian parameters (median $r = 0.82$), hinting to a role of the

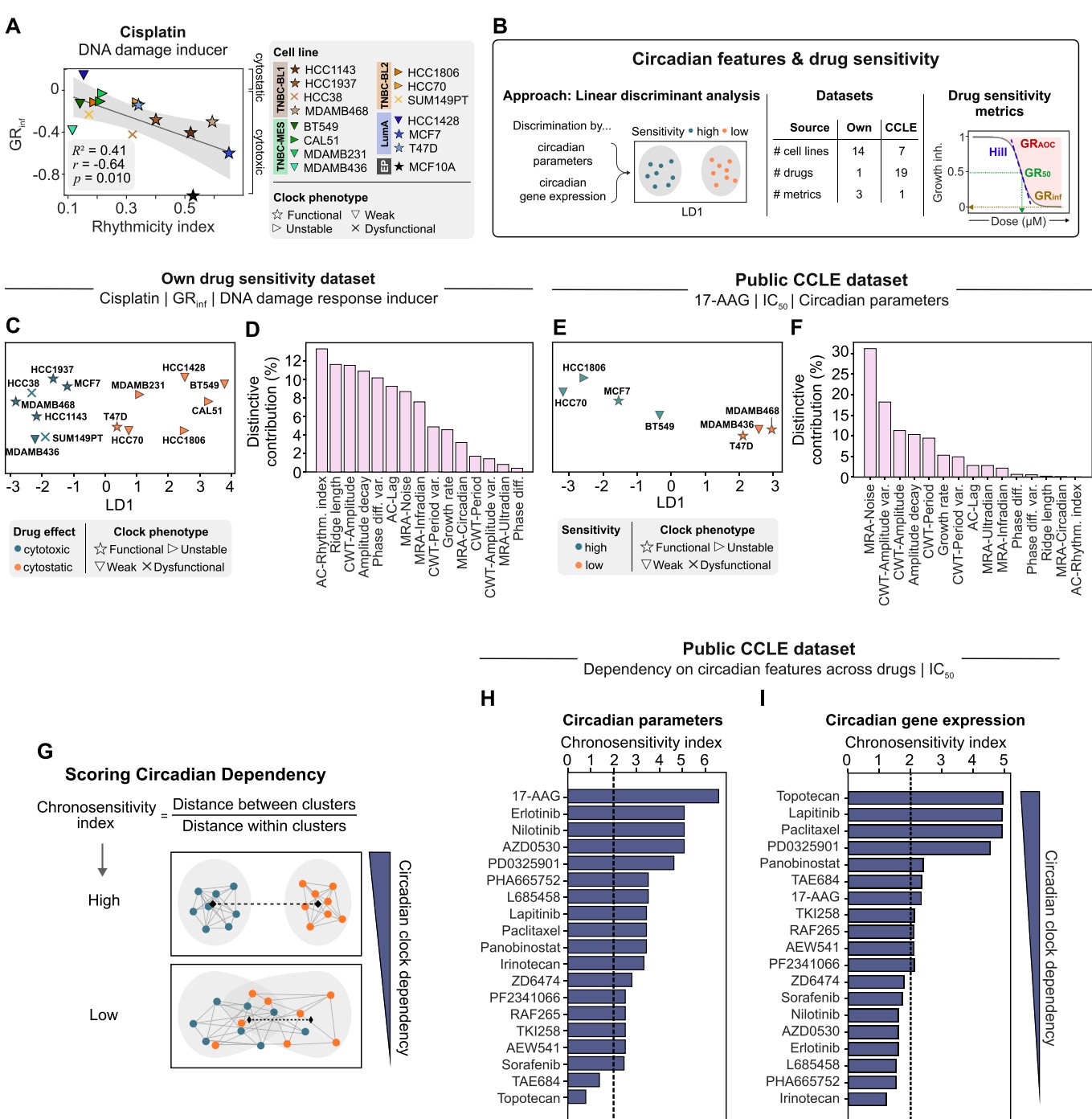

circadian clock in driving response phenotypes to the PI3K inhibitor (Fig. EV3B). Vice versa, we found the highest overall correlation for the circadian amplitude parameter (median $r = 0.74$), suggesting a predictive role for maximal drug responses (Fig. EV3C). This was supported by significant correlations between circadian parameters and other drug sensitivity metrics, like the drug concentration at half-maximal effect ($GR_{50}$) (Fig. EV3D–F), and the slope of the dose-response curve (Hill coefficient) (Fig. EV3G–I).

Despite identifying selected significant correlations between individual clock or growth parameters and drug responses, overall correlations were modest. Given the multilayered nature of the circadian clock, drug sensitivity is likely influenced not by a single circadian factor, but by the collective impact of various circadian features. To investigate this, we utilized linear discriminant analysis (LDA) on both our own and publicly available drug sensitivity datasets, enabling the screening of various drug classes (Fig. 4B; information about drugs listed in Dataset EV2). LDA not only

Figure 4. Investigating the role of circadian features in driving drug sensitivity.

(A) Linear regression analysis between cisplatin $GR_{inf}$ values (absolute drug effect; annotated on the right side of the plot) and the rhythmicity indices assessed by autocorrelation analysis (averaged *Bmal1-Per2* data, see Fig. 2B for exact samples size). Cell models are color-coded and illustrated with their circadian phenotype by distinct markers. Gray-shaded area = 95% CI. Model accuracy indicated by $R^2$-values. r, Pearson correlation coefficient, p, Pearson correlation p-value. (B) LDA-based approach and analyzed datasets to study the role of circadian features in shaping drug sensitivity profiles. Sketch on the right illustrates a dose–response curve and corresponding drug sensitivity parameters that were analyzed. (C) LDA on median-binarized $GR_{inf}$ values to cisplatin, using *Bmal1-Per2* circadian and growth parameters as predictors (n = 15 parameters). Cell models exhibiting drug sensitivity values below or above the cell panel's median are colored in blue-green and orange, respectively. The x-axis represents the first linear discriminant (LD1), capturing the maximum variance between groups. (D) Bar plot ranking circadian and growth parameters by their respective contribution to the obtained discriminative information shown in (C). (E) LDA on median-binarized $IC_{50}$ values to 17-AAG. (F) Bar plot ranking the contribution of each parameter to the obtained discriminative information shown in (E). (G) Approach to identify circadian dependencies of drugs (termed as "chronosensitivity index"), calculated from the between-cluster-distance (BCD) to within-cluster-distance (WCD) ratio. The higher the chronosensitivity index, the more accurate the distinction into drug sensitivity groups, hence the higher the dependency on the circadian clock. (H) Ranking of chronosensitivity indices of multiple drugs, based on their dependency on circadian oscillation parameters. (I) Ranking of chronosensitivity indices of multiple drugs, based on their dependency on circadian gene expression values. The dashed line (y = 2) denotes the cut-off for good and poor discrimination within the LDA space. Source data are available online for this figure.

assesses the individual contributions of the parameters to the classification into drug sensitivity groups but also facilitates the identification of drugs whose effects are likely driven by circadian rhythms (see Methods).

In our dataset, LDA successfully differentiated cell models based on their cytostatic or cytotoxic responses to cisplatin (Fig. 4C). The circadian parameters contributed to varying degrees to this discrimination, with the rhythmicity index and ridge length collectively explaining ~25% of the separation between the classes, emerging as the most influential discriminative features (Fig. 4D). To deepen our understanding of the molecular mechanisms by which the circadian clock modulates cellular responses to drug perturbations, we next investigated the influence of circadian clock gene expression on drug responses. In line with our findings for the circadian parameters obtained from our deep circadian phenotyping approach, the circadian gene set was able to broadly categorize cell models by their responses to cisplatin (Fig. EV4A, left panel). Examining the individual linear discriminant components of each circadian clock gene revealed a clear ranking of genes in terms of their discriminative importance, with the core clock genes *Rorα* and *Cry2* being the most important contributors, collectively accounting for approximately 34% of the discriminative information (Fig. EV4A, right panel). The potential of circadian parameters, growth rates and circadian genes to reliably group cell models according to their sensitivity to cisplatin was further identified for $GR_{50}$ values and Hill coefficients of the response curves (Fig. EV4B–E).

Given the sample size constraints in our study, we primarily used LDA for sensitivity classification. To explore alternative methods for assessing the cumulative impact of circadian features on drug sensitivity, we also evaluated ridge and lasso regression models, preserving continuous drug sensitivity values, and logistic regression as an additional classification method. Accuracy of each method was measured as relative between true and predicted values for ridge and lasso regression, and by leave-one-out cross-validation for the classification methods, using multiple sensitivity metrics, including $GR_{AOC}$ and $GEC_{50}$ (Hafner et al, 2016). This revealed moderate to low mean accuracies for all methods, likely reflecting limited sample size and varying dependencies on cell line models and sensitivity metrics (Appendix Fig. S4). Classification-based approaches (LDA and logistic regression) achieved mean accuracies of 0.42–0.51, outperforming regression methods, which showed up to two-fold lower accuracies (Appendix Fig. S4A–D).

This validation analysis reinforces confidence in our LDA-based approach while highlighting significant variability across cell lines and drug sensitivity parameters in their predictability.

To broaden the scope of our analysis, we incorporated publicly available drug sensitivity data from a subset of our circadian-phenotyped cell models. Our analysis revealed that sensitivity to the HSP90 inhibitor 17-AAG was particularly well discerned by circadian parameters (Fig. 4E). Among these, factors related to clock stability contributed most to the classification of cell models into high and low sensitivity groups, based on their respective $IC_{50}$ values (Fig. 4F). To quantitatively evaluate LDA performances and the discriminative power of circadian features on drug responses, we calculated the ratio of between-cluster-distances (BCD) to within-cluster-distances (WCD), a standard metric for assessing clustering separation (Tibshirani et al, 2002). We termed this ratio the "chronosensitivity index" of a drug (Fig. 4G), to highlight the context of our analysis and make the ratio more interpretable. Strikingly, the circadian parameters provided a well separation of cell models into sensitivity groups for almost all 19 drugs analyzed (Fig. 4H), as indicated by chronosensitivity indices greater than 2, which we defined as a threshold for effective separation (see Fig. EV4F for a direct comparison of LDA profiles with different chronosensitivity indices). Consistent with its high chronosensitivity index, 17-AAG demonstrated the highest accuracy in LOOCV analysis, correctly classifying 71% of cell lines, while the lowest-ranking drug, topotecan, showed the poorest performance with only 14% accuracy (Appendix Fig. S5A). Although the LDA accuracy for other top-ranked drugs, including erlotinib, nilotinib, and AZD0530, was lower, complementary LOOCV using logistic regression revealed an accuracy of 71% for these drugs, which supports their elevated chronosensitivity rankings (Appendix Fig. S5A,B).

Expanding on these findings, circadian clock gene expression also demonstrated a strong association with sensitivity patterns for several drugs based on their chronosensitivity index (Fig. 4I). However, for approximately half of the drugs, the chronosensitivity index fell below the threshold, indicating poor separation into sensitivity groups. Notably, while circadian parameters alone did not effectively classify cell models by their sensitivity to the topoisomerase inhibitor topotecan, its chronosensitivity index and classification performance were substantially improved when using the circadian gene expression panel (Fig. 4I; Appendix Fig. S5C). Interestingly, the top three scoring drugs in the chronosensitivity

index derived from circadian gene expression data, namely topotecan, lapatinib, and paclitaxel, were with 71% to 86% accuracy in LOOCV for LDA also the most robustly classified drugs (Fig. 4I; Appendix Fig. S5C), although such accuracy was not observed in logistic regression analysis (Appendix Fig. S5D,D).

In summary, our analysis underscores the circadian clock's critical role in determining drug sensitivity across a wide range of drugs. We demonstrate that a combination of circadian features, rather than a single factor, discriminates between high and low drug responses. Through the "chronosensitivity index", we introduce a metric for quantifying the circadian clock's impact on differentiating between high and low drug sensitivity groups, thereby identifying potential drugs that act in a circadian-dependent manner.

# Discussion

Despite the wide acknowledgement of the fundamental role that the circadian clock has in the progression of diseases and in the modulation of treatment responses (Lévi et al, 2010; Mormont and Levi, 2003; Sulli et al, 2019), our understanding of the underlying molecular mechanisms remains fragmented. Using a high-resolution circadian phenotyping approach to thoroughly characterize circadian rhythms in cancer models, we discovered strong circadian rhythms in numerous breast cancer cell models. Among these, models of the most aggressive TNBC subtype showed distinct circadian rhythms, challenging the general expectations that the circadian clock is rather dysregulated in highly transformed cancer (Mormont and Levi, 2003; Ye et al, 2018) (Fig. 2). Our measurements of the circadian clock in the non-malignant epithelial cell line MCF10A, luminal A (LumA) breast cancer cell line MCF7, and the osteosarcoma U-2 OS cells aligns with prior research (Baggs et al, 2009; Börding et al, 2019; Lellupitiyage Don et al, 2019; Lin et al, 2019), proving the applicability and flexibility of our approach for deep circadian phenotyping across tissue and cancer types.

In our study, we identified a range of *Bmal1-Per2* phase differences across and within various tissue types, where non-cancerous and osteosarcoma tissues exhibited consistent phase differences around $2\pi/3$ (an 8-h lag in a 24-h cycle), whereas breast cancer subtypes generally showed minimal phase differences ($2\pi$), though with considerable variability (Fig. 2E,F). These observations contrast with the findings of other studies, which identified conserved circadian phase relationships in vivo across different tissues in baboon, mice, and human samples (Mure et al, 2018; Talamanca et al, 2023; Zhang et al, 2014). Those studies utilized intact biological systems, allowing for the influence of systemic cues and the coordination of circadian rhythms across the organism. In contrast, our analysis was conducted on isolated cell lines in vitro, which lack these systemic signals. This methodological difference, along with our examination of various disease subtype models, likely contributes to the broader spectrum of phase differences we observed, particularly among breast cancer subtypes.

Recent work revealed subtype-dependent clock functionality in breast cancer and that breast cancer clocks are critically regulated by estrogen responsiveness (Li et al, 2024). Our analysis of multiple cell models per breast cancer subtype and our broad set of circadian parameters confirms variations in clock strength across breast

cancer subtypes, however, we also revealed substantial variability within the individual subtypes of breast cancer. This variety of distinct clock features motivated us to define circadian clock-based subtypes within the panel of cell lines tested. In detail, we systematically categorized cell models into groups with functional, weak, unstable, or dysfunctional clocks, based on representative parameters indicative of circadian strength, stability, and periodicity (Fig. 3C). This classification diverged from traditional subtyping, exposing functional clocks in only two of the three tested estrogen-positive LumA cell models, alongside three out of four estrogen-negative TNBC models. Together, the defined clock subtypes offer a complementary viewpoint on the subtyping of breast cancer which could refine the assessment of cancer models for circadian-aligned therapeutic strategies and improve the prediction of their outcomes. The previously mentioned study on assessing global rhythm strength in breast cancer patient samples following an informatic approach emphasizes the importance of tumor subtype-specific analysis of circadian rhythms (Li et al, 2024). Our findings further support this need and highlight the importance of assessments that consider the distinct characteristics of each tumor individually.

On the molecular level, circadian clocks ensure coordinated rhythmicity from the individual cell to whole organism level by organized TFFLs, regulated through the interaction of multiple circadian clock genes (Chaix et al, 2016; Takahashi, 2017). The functionality of the circadian clock is highly dependent on genomic integrity, as proved by animal studies revealing distinct phenotypic effects for mutations of mammalian clock genes (Ko and Takahashi, 2006). When exploring the potential to infer our established circadian-based subtypes from publicly available genetic profiles of the cell models, we found high mutation burdens in circadian genes and compromised clock functionality within the unstable and weak circadian subtypes, which suggests a genetic basis for circadian disruption in these groups. However, for dysfunctional clocks, the mutation burden was not significantly higher compared to functional clocks, indicating that factors beyond mutation burden may influence clock dysfunction (Fig. 3E). Here, clock dysfunction may be driven by non-genetic factors, including epigenetic modifications or post-translational modifications of clock proteins, which do not necessarily correlate with increased mutation burdens (Liu et al, 2024).

Illuminating the link between the circadian clock and drug sensitivity, we found strong correlations between selected circadian features and drug sensitivity metrics (Fig. EV3). Analyzing absolute correlation values for each circadian metric, we identified circadian amplitude as the most correlated metric with both maximal drug responses ($GR_{inf}$) and the concentration at half-maximal drug growth inhibition ($GR_{50}$) across nine distinct drugs. However, despite these significant correlations, overall pairwise correlation strength was fairly moderate. This suggests that drug sensitivity is not shaped by a single circadian factor, but rather by the collective influence of multiple circadian features which we confirmed by linear discriminant analysis. Furthermore, we found that the most distinctive circadian features vary depending on the specific drug or sensitivity parameter in question. For example, while strength-related circadian parameters such as the rhythmicity index and circadian amplitude were strong predictors for the drug sensitivity to the DDR-inducer cisplatin (Fig. 4D), parameters indicative for clock stability were mostly contributing to the discrimination into sensitivity groups to the HSP90 inhibitor 17-AAG (Fig. 4F). These results suggest a complex

relationship between circadian rhythms and drug responses that needs to be evaluated on a drug-by-drug basis.

With the "chronosensitivity index" we systematically quantified the dependency of drug responses on the circadian clock, based on the ability of circadian parameters or circadian gene expression levels to effectively classify cell models into distinct sensitivity groups. By that we identified circadian clock-sensitivity relationships for most drugs studied, highlighting a global influence of the circadian system on pharmacodynamics (Fig. 4H,I). Importantly, 17-AAG ranked highest in the chronosensitivity index and predictive accuracy for separating the cell line panel into high- and low-response groups based on circadian clock parameters. High-throughput screenings of time-of-day effects on anticancer drug activity by Lee et al, revealed high time-of-day-specific action for the HSP90 inhibitor 17-AAG, mediated by the circadian regulation of the cell cycle (Lee et al, 2021). Our time-of-day-independent drug sensitivity results complement these findings, underscoring a critical impact of circadian rhythms on drug efficacy and highlighting the necessity of incorporating circadian biology into pharmacological research and treatment strategies. Though, while circadian parameters and gene expression profiles proved to offer insights into drug responses, differences in their predictive power for certain drugs, such as topotecan, suggest areas for future research.

It is important to note that variations in growth rate among cell lines could potentially influence bioluminescence reporter readings, independently affecting perceived circadian rhythms. To ensure that observed effects are specifically attributable to circadian regulation rather than confounding factors like cell cycle dynamics (Bieler et al, 2014; Gutu et al, 2024), future studies could employ single-cell imaging techniques where both the circadian clock and the proliferation behavior of individual cells is measured. This approach would enable more precise measurements of circadian rhythms, reducing potential interference from high proliferation rates during bioluminescence recordings. Moreover, the drugs identified in this study to likely depend on circadian clock properties could be tested in single-cell time-of-day treatment settings, which could also help identify potential bidirectional drug effects on the circadian clock (Manella et al, 2021).

While our study provides valuable insights into circadian rhythm variations in breast cancer cell models, it is crucial to acknowledge the limitations in transferring these findings directly to a clinical setting. Recent work by the Anafi/Meng groups utilizing patient-derived models suggests alternative approaches that may bridge this gap by more closely mimicking the in vivo environment (Anafi et al, 2017). Therefore, while our results contribute to a deeper understanding of circadian dynamics in breast cancer cell lines, extrapolation to patient tumors requires cautious interpretation and further validation using patient-derived models.

In summary, the proposed methodological refinements and follow-up studies offer promising opportunities to deepen our understanding of circadian rhythms in breast cancer. By illuminating the role of the circadian clock across various breast cancer cell models, our research establishes a foundational framework for exploring chronotherapeutic strategies. These insights potentially pave the way for more personalized and circadian-based therapeutic interventions, which could transform our approach to breast cancer treatment.

# Methods

**Reagents and tools table**

| Reagent/Resource | Reference or Source | Identifier or Catalog Number |
| --- | --- | --- |
| **Experimental models** | | |
| HCC1143 | ATCC | CRL-2321 |
| HCC1806 | ATCC | CRL-2335 |
| HCC1937 | ATCC | CRL-2336 |
| HCC38 | ATCC | CRL-2314 |
| HCC70 | ATCC | CRL-2315 |
| MDAMB468 | ATCC | HTB-132 |
| BT549 | Gift of Prof. Peter Sorger (Harvard) | |
| CAL51 | Gift of Prof. Peter Sorger (Harvard) | |
| HCC1428 | Gift of Prof. Peter Sorger (Harvard) | |
| MDAMB231 | Gift of Prof. Peter Sorger (Harvard) | |
| MDAMB436 | Gift of Prof. Peter Sorger (Harvard) | |
| SUM149PT | Gift of Prof. Peter Sorger (Harvard) | |
| MCF10A | Gift of Prof. Joan Brugge (Harvard) | |
| MCF7 | Gift of Prof. Joan Brugge (Harvard) | |
| T47D | Gift of Prof. Joan Brugge (Harvard) | |
| U-2 OS | Gift of Prof. Achim Kramer (Charité) | |
| U-2 OS Cry1-sKO | Börding et al, 2019 | |
| U-2 OS Cry2-sKO | Börding et al, 2019 | |
| U-2 OS Cry1/2-dKO | Börding et al, 2019 | |
| HEK293T | Gift of Prof. Galit Lahav (Harvard) | |
| **Recombinant DNA** | | |
| pAB-mBmal1:Luc-Puro | Gift of Prof. Achim Kramer (Charité) | |
| plenti6-mPer2:Luc-Blas | Gift of Prof. Achim Kramer (Charité) | |
| psPAX2 | Addgene | Cat #12260 |
| pMD2G | Addgene | Cat #12259 |
| gag/pol packaging plasmid | Addgene | Cat #14887 |
| pRev packaging plasmid | Addgene | Cat #12253 |
| VSV-G envelope plasmid | Addgene | Cat #14888 |
| EF1α-mKate2-NLS plasmid | Gift of Prof. Galit Lahav (Harvard) | |

| Reagent/Resource | Reference or Source | Identifier or Catalog Number |
|---|---|---|
| **Antibodies** | | |
| Not applicable | | |
| **Oligonucleotides and other sequence-based reagents** | | |
| Not applicable | | |
| **Chemicals, Enzymes and other reagents** | | |
| DMEM/F12 | Gibco | Cat #11320033 |
| RPMI-1640 | Gibco | Cat #21875091 |
| FluoroBrite DMEM | Gibco | Cat #A1896701 |
| Horse serum | Gibco | Cat #26050088 |
| Fetal bovine serum | Gibco | Cat #10270106 |
| EGF Peprotech | Gibco | Cat #AF-100-15-500UG |
| Hydrocortisone | Sigma | Cat #H-0888-1g |
| Cholera Toxin | Sigma | Cat #C-8052-2mg |
| Insulin | Sigma | Cat #I-1882 |
| Penicillin-Streptomycin | Gibco | Cat #15140122 |
| L-Glutamine | Gibco | Cat #25030-024 |
| HEPES | Gibco | Cat #15630056 |
| Lipofectamine 3000 | Invitrogen | Cat #L3000001 |
| Protamine sulfate | Sigma | Cat #P3369 |
| PBS | Gibco | Cat #10010056 |
| Blasticidin | Adooq | Cat #A14212 |
| Puromycin | Gibco | Cat #A1113803 |
| Dexamethasone | Sigma | Cat #D4902 |
| D-Luciferin | Abmole | Cat #M9053 |
| 5-Fluorouracil | Sigma | Cat #03738-100MG |
| Torin2 | Sigma | Cat #SML1224-5MG |
| Alpelisib | Biozol | Cat #TGM-T1921-10MG |
| Adavosertib | Biocat | Cat #T2077-5mg-TM |
| Doxorubicin | Hölzel | Cat #A14403-100 |
| Paclitaxel | Hölzel | Cat #M1970-50mg |
| Alisertib | Hölzel | Cat #S1133-5 |
| Olaparib | Adooq | Cat #A10111 |
| Cisplatin | Sigma | Cat #232120 |
| **Software** | | |
| MATLAB v2021b | https://mathworks.com | |
| Python Spyder v5.4.5 | https://github.com/spyder-ide/ | |
| pyBOAT v0.9.12 | Mönke et al, 2020 | |
| Anaconda Navigator v2.5.0 | https://docs.anaconda.com/navigator/ | |
| **Other** | | |
| LumiCycle | Actimetrics | |
| Incucyte | Essen BioScience | |

## Experimental methods

### Cell lines and cell culture

MCF10A cells were cultured following the Brugge lab's media recipe including a DMEM/F12 medium base (Gibco) supplemented with 5% horse serum (Gibco), 20 ng/ml EGF (Peprotech), 0.5 mg/ml Hydro-cortisone (Sigma), 100 ng/ml Cholera Toxin (Sigma), 10 µg/ml Insulin (Sigma) and 1% penicillin-streptomycin (Pen-Strep, Gibco). All other cell lines were maintained in RPMI-1640 medium (Gibco) supplemented with 10% fetal bovine serum (FBS) (Gibco) and 1% Pen-Strep. Bioluminescence recordings and live imaging required phenol red-free FluoroBrite DMEM medium, supplemented with 10% FBS, 300 mg L-Glutamine (Gibco) and 1% Pen-Strep. Cells were cultured in a controlled 37 °C and 5% $CO_2$ environment.

### Cell line quality control

All cell lines were monitored for morphology, growth characteristics and health, mostly by long-term live-cell imaging, confirming that cell line-specific features did not vary throughout the study. No commonly misidentified cell line, according to the ICLAC register, was used in this study. Cells were routinely tested for mycoplasma for quality assurance.

### Generation of luciferase and fluorescence reporter cell lines

HEK293T cells at 80% confluency were retained in RPMI-1640 medium supplemented with 10 mM HEPES and transfected with a mix of 8.4 µg lentiviral expression plasmid (pAB-mBmal1:Luc-Puro or plenti6-mPer2:Luc-Blas), 6 µg psPAX2 (Addgene #12260) and 3.6 µg pMD2G (Addgene #12259) to produce lentivirus carrying circadian luciferase reporters. Similarly, lentivirus for the red-fluorescent nuclear reporters were produced using a mix of 1.8 µg gag/pol packaging plasmid (Addgene #14887), 0.7 µg pRev packaging plasmid (Addgene #12253), 0.3 µg VSV-G envelope plasmid (Addgene #14888) and 3.2 µg of EF1α-mKate2-NLS plasmid. The transfections were performed using Lipofectamine 3000 (Invitrogen) according to the manufacturer's instructions. Virus was harvested and filtered through a 0.45 µm filters (Millipore) 48 h and 72 h post-transfection. For transduction, target cells at 70% confluence underwent a 6-h incubation with a mix of 1 ml lentivirus-containing supernatant, 8 µg/ml protamine sulfate (Sigma) and 10 µM HEPES (Gibco). Post-incubation, cells were washed with PBS (Gibco) and maintained in standard culture medium. After 2 days, antibiotic selection of transduced cells was initiated using either 5 µg/ml of blasticidin (Adooq) or 2 µg/ml of puromycin (Gibco), depending on the resistance marker present in the lentiviral expression vector. Details on how the different U-2 OS knockout cell lines were developed are described in the original publication (Börding et al, 2019).

### Circadian bioluminescence recordings

Cells expressing luciferase reporters under the control of the *Bmal1* or *Per2* promoters were plated in 35-mm dishes (Nunc) to reach approximate confluence by the next day. To synchronize single-cell circadian clocks in cell populations, a standard dose of 1 µM dexamethasone (Balsalobre et al, 2000; Finger et al, 2021) (Sigma, dissolved in EtOH) was applied. Following a 30-min incubation period, cells were rinsed once with PBS before adding imaging medium containing 250 µM D-Luciferin (Abmole). To prevent the imaging medium from evaporating during the duration of the

bioluminescence measurements, the dishes were sealed with parafilm (Finger et al, 2021). Bioluminescence was then recorded every 10 min for a span of 5.7 days, utilizing an incubator-embedded luminometer (LumiCycle, Actimetrics).

### Live-cell imaging

Continuous live-cell monitoring was performed utilizing an incubator-embedded live-cell widefield microscope (Incucyte, Essen BioScience). Cells expressing the fluorescent nuclear marker EF1α-mKate2-NLS were plated in 48-well plates (Falcon) at a seeding density where cell growth saturated under normal growth conditions by the conclusion of each experiment. Growth was measured in both brightfield and red-fluorescent channels with an excitation between 567 and 607 nm and emission between 622 and 704 nm. The acquired images were subjected to per-frame analysis for nuclei counting using the integrated software of the Incucyte system, followed by data processing in MATLAB. In experiments involving pharmacological treatments, cell seeding was done 1 day prior to the initiation of imaging. We employed a 4× objective to take images across two fields of view for each well, at intervals of 1–2 h, over a period of 4 days. These experiments were replicated across two separate plates for each condition. In the case of cisplatin treatment studies, we conducted the experiment in a single plate per condition, capturing images across nine fields of view with a 10× objective. For the assessment of unperturbed growth dynamics, we seeded cells at approximately 10% confluency, two days before initiating the live imaging. Using a 10× lens, we obtained nine snapshots per well at 1 to 2-h intervals, for 4 days.

### Drug perturbations

Stock solutions for drugs were prepared in DMSO and stored at −20 °C, except for cisplatin (Sigma) that was prepared in 0.9% NaCl and kept at room temperature. For dose-response experiments, drugs were freshly diluted in a logarithmic series (5–6 points, log4) using their respective solvents just before application. The following concentration ranges were evaluated: 5-fluorouracil (5-FU, Sigma), alpelisib (Biozol), and olaparib (Adooq) were tested from 100 to 0.1 µM; torin2 (Sigma) and alisertib (Hölzel) from 10 to 0.01 µM; adavosertib (Biocat) from 10 to 0.04 µM; doxorubicin (Hölzel) from 1 to 0.004 µM; and paclitaxel (Hölzel) from 0.4 to 0.0004 µM. Cisplatin underwent a 10-point log2 dilution series, spanning 70 to 0.14 µM. The compounds were mixed media to achieve a final volume of 9% of the well and administered to the cells 24 h post-seeding. Cisplatin doses were given 48 h post-seeding. Control groups received the respective solvent to account for potential solvent-based effects on growth. Growth of the cells was tracked by long-term live-cell imaging over a minimum period of 4 days, as previously detailed.

### Experimental study design and statistics

The sample size for this study was based on standard field practices. We conducted two independent bioluminescence experiments, each with two to three technical replicates. While MCF10A Bmal1- and Per2-Luc and MDAMB468-Per2-Luc studies lacked biological replication, the technical replicates demonstrated high consistency. Given the extensive data generated, live-imaging experiments were performed once, with nine images analyzed per cell line for growth and cisplatin sensitivity analysis, and four images per condition from two separate plates for sensitivity analysis involving other drugs. Due to the study design not involving large treatment and control cohorts, we did not employ randomization procedures for sample allocation, nor did we implement blinding during experiments or data analysis. Data points were included in the analysis unless identified as outliers using box plot methods, with pre-established exclusion criteria of 1.5 times the interquartile range above the 75th or below the 25th percentile. Statistical analyses were tailored to the nature of the data in each figure. For cluster analysis, we used k-means clustering and determined optimal cluster numbers through elbow plots, validating our approach with Within-Cluster Sum of Squares (WCSS) and silhouette scores. We assessed statistical parameter relationships using Pearson correlation coefficients and their corresponding p-values. The LDA model's performance was evaluated using the BCD to WCD ratio and cross-validated through LOOCV. Throughout the analysis, group variability was estimated and reported as standard deviation.

# Computational methods

## Time-series analysis of circadian signals

### Signal pre-processing

An initial dexamethasone-related signal peak within the first 5 h was uniformly removed from all raw time-series. To standardize the duration of recordings, time-series exceeding 137.7 h (~5.7 days) were shortened accordingly. In the case of the MDAMB468 Per2-Luc reporter cell line, one replicate required a further adjustment by shortening the dataset by 8 h (129.7 h). This was due to an anomalous, intense, and brief peak during the last recording hours that skewed the analysis of circadian parameters, and that was not present in other replicates. The signals were detrended with a 48-h cut-off period using a sinc filter. Signal normalization was achieved by inverting the continuous amplitude envelope of the detrended signal (Mönke et al, 2020). The amplitude envelope was calculated by continuous wavelet transform with a time window of 48 h. The signal pre-processing steps were performed in the open-source software package pyBOAT (Mönke et al, 2020) (version 0.9.12), accessed via the Anaconda Navigator (version 2.5.0).

### Autocorrelation analysis

Autocorrelation of detrended signals was calculated using the 'autocorr' MATLAB function. Rhythmicity strength was determined from autocorrelation values at the second peak in the correlogram using the 'findpeaks' MATLAB function, with the corresponding abscissa (lag) reflecting the main period. To facilitate peak detection and to filter out non-periodic samples from further analysis, autocorrelation values were smoothed using a Gaussian filter and peaks outside the 95.5% confidence intervals or below 0 were not considered for further analysis.

### Multiresolution analysis

Decomposition of detrended signals into the underlying frequency bands, referred to as wavelet details $D_j$, was done by a discrete wavelet transform-based multiresolution analysis (Leise and Harrington, 2011), ensued by a final smooth. The analysis was carried out in accordance with the method outlined by Myung, Schmal and Hong et al (Myung et al, 2018), employing the 'PyWavelets' Python library and utilizing a db20 wavelet from the Daubechies family for the transformation. For each wavelet detail

$D_j$, which encapsulates a frequency span from $2^j\Delta t$ and $2^{j+1}\Delta t$ (where j = 1, 2, 3, …), the time series was resampled by reducing the sampling interval from $\Delta t = 10$ min to $\Delta t = 30$ to capture a circadian frequency range between 16 and 32 h (Leise, 2013).

### Continuous wavelet transform

Continuous wavelet transform (CWT) was employed for the identification of a time-series signal's main oscillatory feature, referred to as ridge, across the entire recording time. The application of this wavelet-based spectral analysis varied with the specific readout parameter being examined. To increase the detectability of oscillatory patterns associated with the "period" and "phase" readout parameters, the CWT was applied to detrended amplitude-normalized signals. Furthermore, an adaptive ridge detection threshold was set according to the quarter-maximal spectral power of each signal (Appendix Fig. S1A, right panel). Conversely, for the readout parameters "amplitude" and "ridge length", indicative of circadian strength and robustness, the analysis was conducted on detrended, unnormalized signals with a global ridge detection threshold that results in a broad distribution of ridge lengths across all samples tested and which we set to a quarter of the median half-maximal wavelet power from the aggregate of samples (Appendix Fig. S1A, left panel). Except for the "ridge length" parameter, we focused on analyzing ridges detectable for a minimum duration of 48 h, thereby ensuring the tracking of at least two full circadian cycles for our subsequent analysis. The CWT was performed with pyBOAT using the Python programming language in the Spyder environment (version 5.4.5). The instantaneous phase differences between *Bmal1* and *Per2* was determined using the 'atan2' function, and for visual representation in polar histograms, the 'polarhistogram' function from MATLAB.

### Weighting of CWT parameters

To account for variations in ridge lengths from which CWT period, amplitude, and phase parameters were derived, we weighted the importance of each sample through a sigmoidal fitting method applied to the lengths of the respective ridges (Appendix Fig. S1B), based on the formula:

$$Relative\ weight = \frac{1}{1 + e^{(-0.1(ridge\ length - x_0))}} \quad (1)$$

Here, the steepness of the curve was uniformly set to $-0.1$, while the reflection point of the curve, $x_0$, lied between the highest and lowest ridge length. This method ensured that parameters from longer ridge lengths had a higher relative weight for median and mean calculations, as opposed to lower-weighted parameters from shorter ridge lengths.

### Amplitude envelope decay rate calculation

Decay rates of amplitude envelopes used for later correlation analyses were derived by applying an exponential decay model to detrended signal amplitudes:

$$f(t) = A_0 e^{-\gamma t} \quad (2)$$

where $A_0$ denotes the initial amplitude, $t$ the time, and $\gamma$ the decay rate. Only decay rates from fits where $R^2 \geq 0.6$ were considered for further analysis. For the analysis of circadian coupling strengths, the

model was extended to fit signals to an exponentially decaying sinusoidal oscillation.

$$f(t) = A_0 e^{-\gamma t} \sin\left(\frac{2\pi}{T_s} t + \phi\right) \quad (3)$$

where $T_s$ is the oscillation period, and $\phi$ adjusts for phase shifts. Again, values obtained from fits where $R^2 \geq 0.6$ were used for further analysis.

### Simulating circadian population signals

The intercellular coupling was modeled assuming cells as mean-field coupled Poincaré oscillators. These dynamics were described in Cartesian coordinates $X$ and $Y$ as:

$$X_{i,n+1} = X_{i,n} + \lambda\left(\sqrt{X^2_{i,n} + Y^2_{i,n}} - R_0\right)X_{i,n}$$
$$- \frac{2\pi}{T_i}Y_{i,n} + \frac{\kappa}{2N}\sum_{i=0}^{N}X_{i,n} \quad (4)$$

$$Y_{i,n+1} = Y_{i,n} + \lambda\left(\sqrt{X^2_{i,n} + Y^2_{i,n}} - R_0\right)Y_{i,n}$$
$$+ \frac{2\pi}{T_i}X_{i,n} + \frac{\kappa}{2N}\sum_{i=0}^{N}Y_{i,n} \quad (5)$$

Here, $N$ represents the number of identical oscillators, $\kappa$ the circadian coupling strength, $T_i$ the inherent period, $R_0$ the natural limit-cycle radius, $\lambda$ the correction strength toward the oscillator's natural radius, $n$ the time step, and $i$ the individual oscillator. To simulate the model, we used the Euler's method with a time step of $\Delta t = 0.1$. For simplicity we set $R_0 = \lambda = 1$, as these parameters had no significant impact on the outcome of the experiment. To reenact the in vivo resetting, all oscillators were given similar initial phases by Uniform$(0, \pi/5)$, and periods with a normal distribution Norm$(24, 3)$. Further, we set $N$ to 300 to simulate enough oscillators and to avoid edge case results while keeping the simulation relatively small. Identical oscillators and uniform coupling in the simulations were justified by our assumption of absent spatial correlations in the experiment. The cumulative cell signal was modeled by summing individual oscillator outputs $\left(\frac{\kappa}{2N}\sum_{i=0}^{N}Y_{i,n}\right)$, varying $\kappa \in [0.00001: 0.1]$, with amplitudes scaled by 0.5 for compatibility, while keeping a constant seed to ensure similar distribution of periods for all simulations.

### Statistical analysis

Significance in variations in circadian parameters between knock-out and wild-type U-2 OS cells were assessed using a two-sample $t$-test through the M ATLAB 'ttest2' function, assuming normal distributions and equal variances between groups. Variations were considered significant with corresponding $p$-values $\leq 0.05$.

## Analysis of growth dynamics

Growth data acquired by long-term live-cell imaging was smoothed by robust local regression of weighted linear least squares combined with a 2nd-degree polynomial model, as implemented by the

'rloess' function MATLAB. The exponential growth rate, denoted as $k$, for each time unit $t$ was determined by normalizing the cell counts to their initial values at timepoint zero ($y_0$) and applying an exponential model to the 4-day growth trajectories:

$$\text{Growth}(t) = y_0 e^{(k*t)} \qquad (6)$$

For the BT549 and MDAMB436 cell lines fluorescent reporter lines were not available, which is why growth was quantified based on confluency levels. For all other cell models, nuclear counting was employed to calculate growth rates. Fitting of the exponential model was employed by the 'fit' function in MATLAB.

## Circadian parameter-based phenotype analysis

The relationships between different circadian parameters and growth rates were analyzed by first averaging the circadian parameter values from both Luc-reporters, *Bmal1* and *Per2*, across breast cancer cell lines. Pearson correlation coefficients and corresponding *p*-values were then calculated between these averaged circadian and growth parameters to reduce redundancy within the dataset. From the reduced set of eight circadian parameters and the growth rate, unsupervised dimensionality reduction by principal component analysis (PCA) was executed using the 'PCA' function of the 'sklearn decomposition' Python language module with the exact full singular value decomposition solver and two output components. Min–max scaling of the circadian parameters was applied prior to processing to preserve the distribution of the data while ensuring that all features are bound between 0 and 1, regardless of their different units and ranges. The reduced 2-dimensional space of *Bmal1-Per2* circadian features of each cell line was plotted in a biplot together with each feature's loading by annotated arrows, revealing three distinct clusters. The resulting component loadings were plotted separately in bars to facilitate the interpretation of each circadian feature's contribution. Based on the feature contributions, one representative parameter was selected from each of the three PCA clusters (period, phase difference variability, circadian component) and their min–max scaled values were plotted in a three-dimensional space. Elbow plots were generated to determine the optimal number of clusters for subsequent k-means clustering, based on the trade-off between cluster coherence and model complexity. We used the KneeLocator python library to locate the most adequate number of clusters (k = 4) with the sensitivity parameter set to 1. The analysis evaluated the average intra-cluster distances across a range of number of clusters ($2 \le n \le 7$).

## Circadian gene expression and mutation analysis

Gene expression data and somatic mutation information of circadian clock genes were sourced from the Cancer Cell Line Encyclopedia Dependency Map (CCLE DepMap, available at https://sites.broadinstitute.org/ccle/datasets, gene expression: 2022-Q2, mutation profiles: 2023-Q2) (Barretina et al, 2012). Cell models were grouped together based on their expression values of core circadian clock genes using the 'seaborn' library's 'clustermap' function, utilizing Euclidean distance for measurement and the complete linkage approach for clustering. We extended our mutation analysis from the set of 16 core clock genes to additional 44 circadian clock associated genes (see Dataset EV1 for gene lists). Cell lines were hierarchically clustered by their mutational burden ranging from zero to greater or equal two, using the hamming

distance metric and single linkage method. Depending on the severity of the mutation, each mutation was assigned a value between 1 to 3 (with 3 reflecting damaging mutations), and final mutation scores per clock phenotype or classical subtype were calculated from averaging individual mutation scores across cell line models.

## Circadian feature and gene expression mapping

Min–max scaled gene expression values of sixteen core clock genes across TNBC cell line models were clustered using the 'clustermap' function of the 'seaborn' Python language module with the Euclidean distance metric and complete linkage method. The Uniform Manifold Approximation and Projection (UMAP) algorithm of the same Python language module was applied on both, the circadian parameters, and circadian gene expression values, to project TNBC cancer cell lines in a two-dimensional space. The number of nearest neighbors selected to construct the initial high-dimensional graph was set to three.

## Estimation of growth rate inhibition values

Drug-response data acquired by long-term live-cell imaging was smoothed using the same approach as described for the growth analysis. Following the method outlined by Hafner et al (2016), the growth rate inhibition ($GR$) at a given time $t$ and for each concentration $c$ was calculated using:

$$GR(c, t) = 2^{k(c,t)/k(0)} - 1 \qquad (7)$$

where $k(c, t)$ represents the growth rate with drug treatment, and $k(0)$ signifies the growth rate of untreated control cells. To obtain final drug sensitivity values, dose-dependent $GR$ values were modeled against a sigmoidal curve, expressed as:

$$GR(c) = GR_{\text{inf}} + \frac{1 - GR_{\text{inf}}}{1 + (c/GEC_{50})^{h_{GR}}} \qquad (8)$$

with the parameters defined as described in the original publication (Hafner et al, 2016). Fitting of the sigmoidal curve was executed with the MATLAB 'fit' function.

## Pairwise correlation and linear regression analysis

Pearson's linear correlation coefficients between drug sensitivity metrics and circadian parameters were calculated and clustered using the 'corr' and 'clustergram' functions in MATLAB, respectively. For clustering, the 'correlation' distance metric was applied. Corresponding linear regression analysis was employed by the MATLAB 'fitlm' function.

## Predicting drug sensitivity from circadian clock features

Drug sensitivity prediction analysis was performed using the supervised linear discriminant analysis algorithm by the 'sklearn discriminant_analysis' Python language module. The default parameters were retained, using the exact full singular value decomposition solver and a single component (LD1). We applied a median-based binary approach to categorize cell models into high and low drug sensitivity groups for cisplatin (own dataset) and 24

additional drugs that we obtained from the CCLE pharmacological profiling dataset archive (2015-Q1) (Barretina et al, 2012). These sensitivity groups served as target variable for the discrimination analysis using either the circadian parameter set (combined *Bmal1-Per2* data), or the CCLE core clock gene expression values. The resulting linear discriminant vector was jittered along the y-axis to avoid overlapping of the data points in the 1-dimensional space. Individual contributions to the obtained discriminative information of the parameters were plotted alongside each linear discriminant plot, and combined sum up to 100%. The discrimination performance ("chronosensitivity index") was calculated from the BCD-WCD ratio, where the between-cluster-distance (BCD) is the Euclidean distance between the mean LD1 values of each class, and the within-cluster-distance (WCD) is the average Euclidean distance of class members to their respective class mean. The minimum BCD-WCD ratio to deem effective discrimination in a single dimension is equal to two, with BCD = 2*WCD. This ensures that the centers of both clusters are at least twice as far away as the distance of each member to its respective class mean.

### Multi-dimensional lasso and ridge regressions

Regression analysis was conducted to identify circadian features associated with drug sensitivity in breast cancer cells. L1 (Lasso) and L2 (Ridge) regularization were applied to reduce overfitting and improve model interpretability by shrinking the coefficients of less significant features to zero (Tibshirani, 1996) (Hoerl and Kennard, 1970). The regularization strengths were set to 0.1 for L1 and 1.0 for L2. All analyses were performed using the scikit-learn library, with default parameters unless otherwise specified.

### Leave-one-out cross-validation

Leave-one-out cross-validation (LOOCV) was employed to evaluate model performance, leveraging the entire dataset by iteratively using one observation as the test set while the remainder constituted the training set. This method provides an unbiased estimate of model performance, particularly for smaller datasets. LOOCV was executed using custom functions with the different types of estimators.

## Data availability

The raw experimental data and generated data tables from this study are available in the Figshare database under the identifier https://figshare.com/projects/Circadian-Subtypes-Breast-Cancer-Cells/234353. The computer code produced in this study is available through the dataset repository Zenodo under the identifier https://doi.org/10.5281/zenodo.14657750.

The source data of this paper are collected in the following database record: biostudies:S-SCDT-10_1038-S44320-025-00092-7.

## Peer review information

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

## Acknowledgements

We would like to express our thankfulness to Nica Gutu for kickstarting the modeling ideas. We also thank Anna-Marie Finger and Astrid Grudziecki for their guidance in the bioluminescence recordings as well as Francesca Müller-Marquardt for her assistance in the recordings. Lastly, we thank the laboratories of Ingeborg Tinhofer-Keilholz and Ulrich Keilholz for their continuous feedback on the project. The results are part of a project funded by the German Federal Ministry of Education and Research (BMBF) through the e:Med Juniorverbund DeepLTNBC TP3-01ZX1917C. CE was partially supported by the Deutsche Forschungsgemeinschaft (DFG, German Research Foundation)–RTG2424/CompCancer – project number: 377984878. JD was supported by the Luxembourg National Research Fund (FNR) under the PRIDE programme (PRIDE17/12252781). MSN acknowledges support from the Novo Nordisk Foundation (NNF20OC0064978). CS acknowledges support from the DFG–SCHM 3362/4–1 project number: 511886499.

## Author contributions

**Carolin Ector**: Conceptualization; Data curation; Software; Formal analysis; Validation; Investigation; Visualization; Writing—original draft; Writing—review and editing. **Jeff Didier**: Data curation; Software; Formal analysis; Validation; Methodology; Writing—review and editing. **Sébastien De Landtsheer**: Data curation; Software; Formal analysis; Validation; Methodology; Writing—review and editing. **Malthe S Nordentoft**: Data curation; Software; Formal analysis; Methodology. **Christoph Schmal**: Software; Methodology; Writing—review and editing. **Ulrich Keilholz**: Resources; Writing—review and editing. **Hanspeter Herzel**: Conceptualization; Writing—review and editing. **Achim Kramer**: Conceptualization; Writing—review and editing. **Thomas Sauter**: Resources; Supervision; Writing—review and editing. **Adrián E Granada**: Conceptualization; Resources; Supervision; Funding acquisition; Visualization; Writing—original draft; Project administration; Writing—review and editing.

Source data underlying figure panels in this paper may have individual authorship assigned. Where available, figure panel/source data authorship is

listed in the following database record: biostudies:S-SCDT-10_1038-S44320-025-00092-7.

## Funding

## Disclosure and competing interests statement

The authors declare no competing interests.

# Expanded View Figures

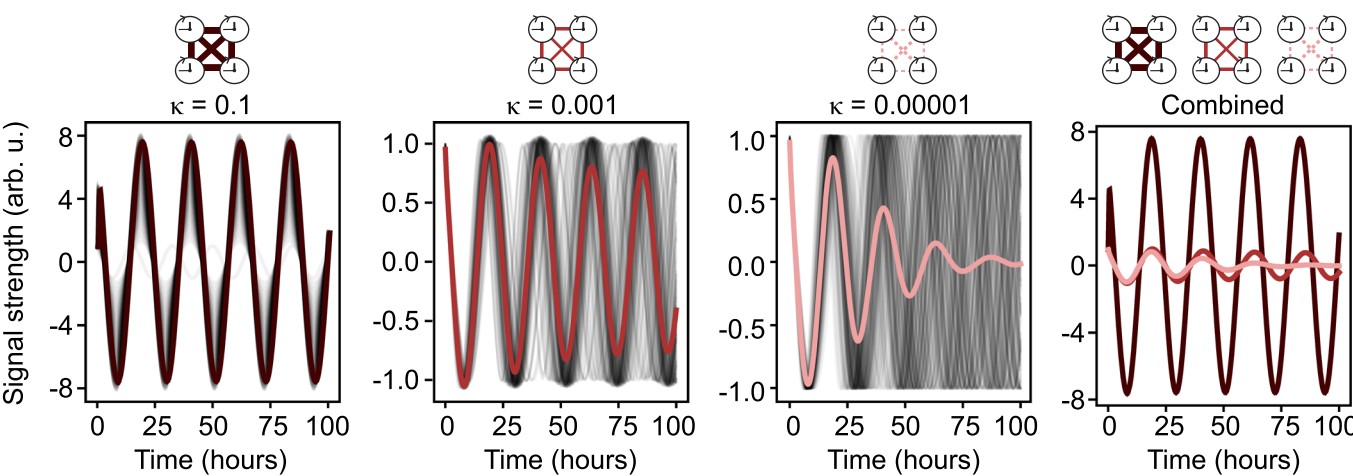

**Figure EV1.   Expanded view for Fig. 2.**

Simulated oscillators of varying coupling strengths κ. Individual traces of each oscillator are shown in black, collective signals are shown as thick lines with gradients of red, based on their coupling strength, where dark red denotes high coupling strength, and pink refers to low coupling. The right panel is a composite of all collective signals.

                                      

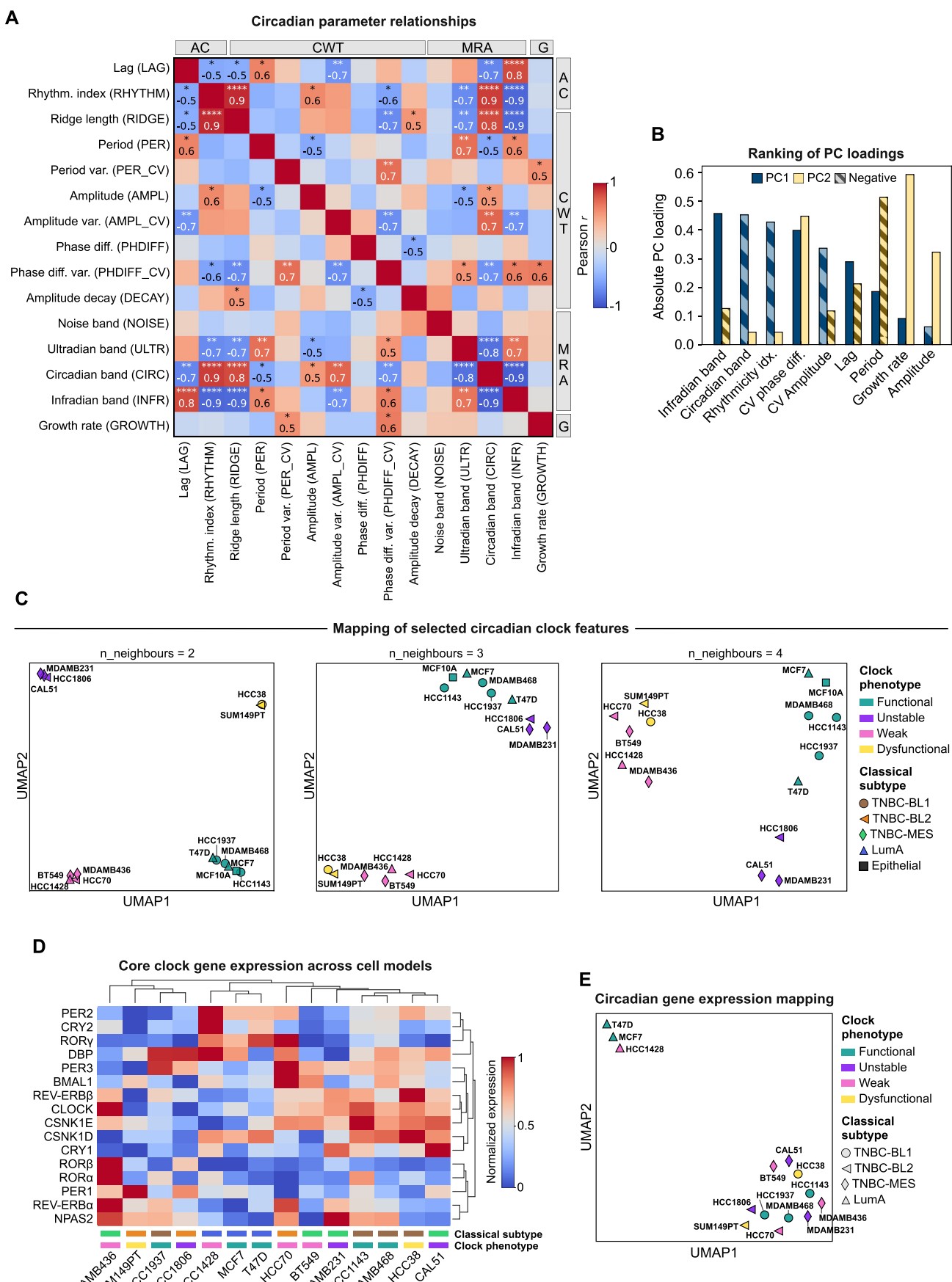

**A** Circadian parameter relationships

**B** Ranking of PC loadings

**C** Mapping of selected circadian clock features

**D** Core clock gene expression across cell models

**E** Circadian gene expression mapping

◀  **Figure EV2.  Expanded view for Fig. 3.**

(A) Pearson correlation coefficients between the complete set of *Bmal1-Per2*-averaged circadian features and growth rates across all breast cancer cell models and the epithelial MCF10A cell line ($n = 15$ cell lines). Parameters are categorized by their approach of calculation (refer to Fig. 3A legend). Displayed are statistically significant correlation values, where *, **, and **** indicate *p*-values < 0.05, 0.01, and 0.0001, respectively. Exact *p*-values: RIDGE-LAG = $4.2 \times 10^{-2}$; RIDGE-RHYTHM = $2.8 \times 10^{-7}$; PHDIFF_RIDGE = $2 \times 10^{-3}$; PHDIFF_CV-PER_CV = $3 \times 10^{-3}$; DECAY-RIDGE = $4.3 \times 10^{-2}$; DECAY-PHDIFF = $3.3 \times 10^{-2}$; ULTR-RHYTHM = $3 \times 10^{-3}$; ULTR-RIDGE = $4 \times 10^{-3}$; ULTR-PER = $3 \times 10^{-3}$; ULTR-AMPL = $3.8 \times 10^{-2}$; ULTR-PHDIFF_CV = $4.1 \times 10^{-2}$; ULTR-INFR = $5.5 \times 10^{-3}$; CIRC-RIDGE = $8 \times 10^{-5}$; CIRC-ULTR = $8.8 \times 10^{-5}$; INFR-RIDGE = $1.8 \times 10^{-5}$; INFR-ULTR = $5.5 \times 10^{-3}$; INFR-CIRC = $1 \times 10^{-6}$; GROWTH-PER_CV = $3.7 \times 10^{-2}$. Refer to Fig. 3A for all other significant *p*-values. (B) Ranking of absolute PC loadings for each circadian and growth parameter, corresponding to the PCA biplot shown in Fig. 3B. Parameters are sorted in descending order based on their absolute contribution in the first principal component. Negative values are shaded. (C) Uniform Manifold Approximation and Projection (UMAP) of cell models complementary to the k-means clustering analysis in Fig. 3C. The classical subtype and clock phenotype of each model is illustrated by different markers and colors, respectively. Nearest neighbors = 2–4. (D) Cluster map of core clock gene expression values across breast cancer cell models, using the Euclidian distance method. Color-coded rectangles above the cell line names indicate the classical subtype and clock phenotype. Refer to (C) for color-coding. (E) UMAP of cell models based on core circadian gene expression values shown in (D). The classical subtype and clock phenotype of each model is illustrated by different markers and colors, respectively. Nearest neighbors = 3.

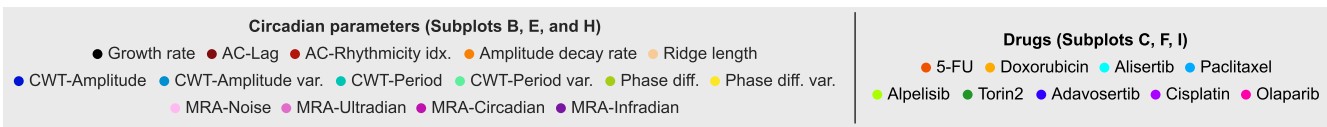

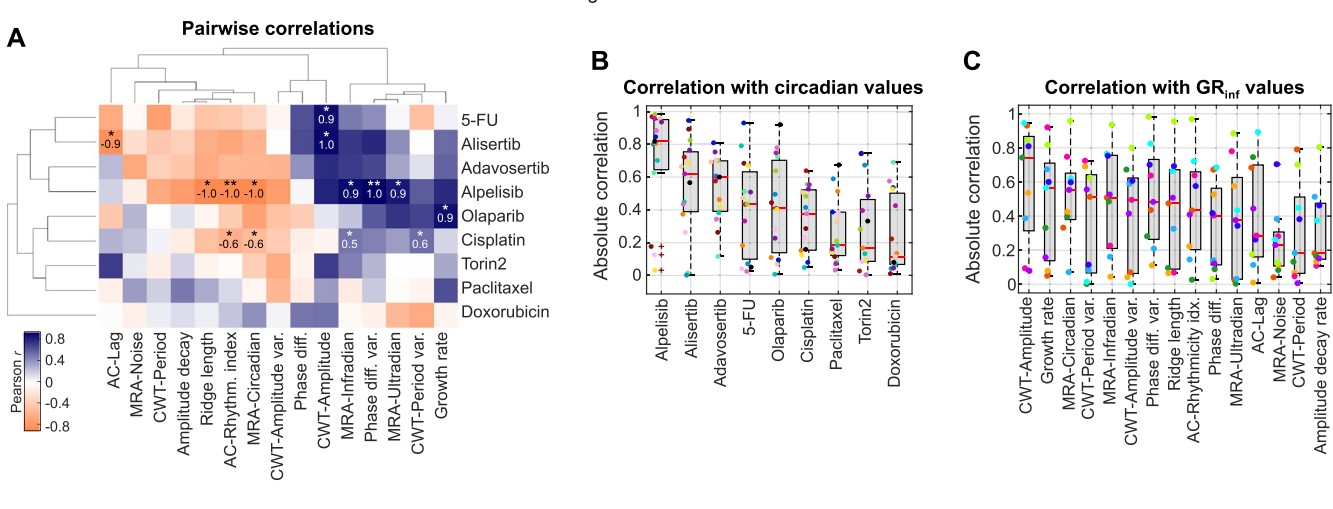

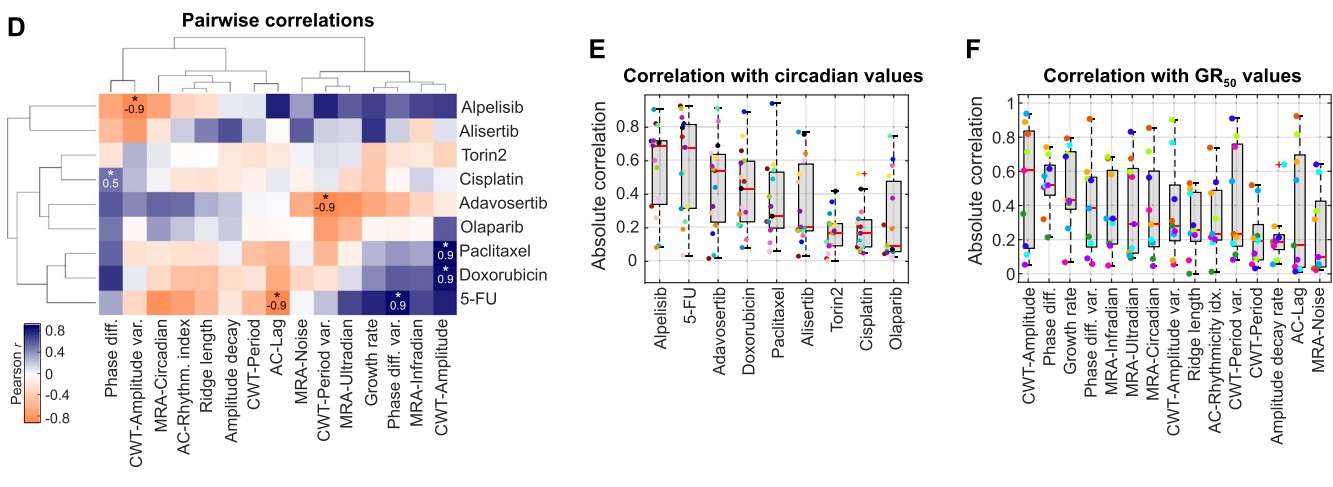

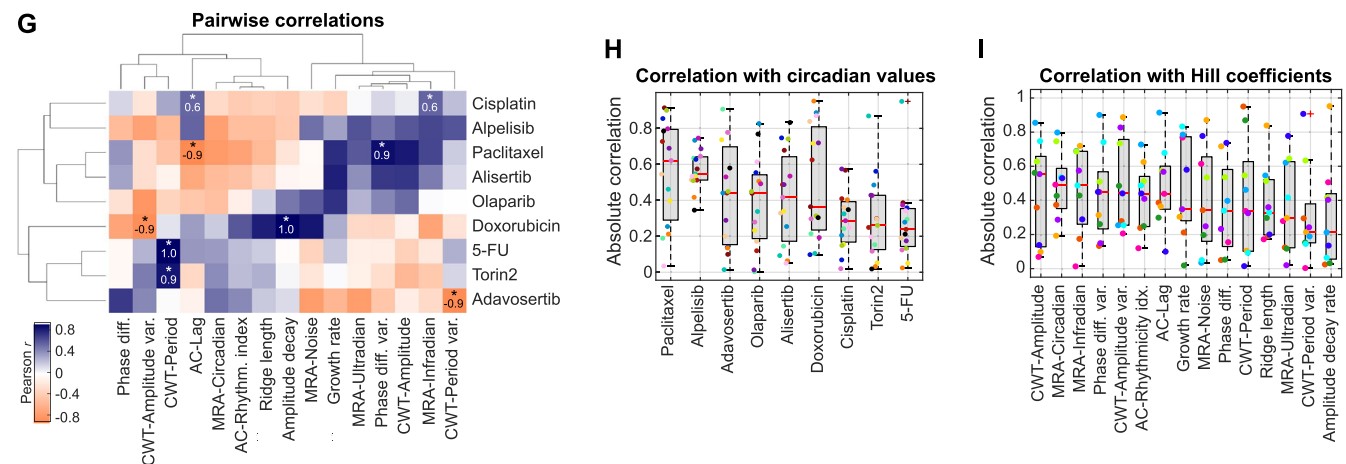

**Figure EV3.  Expanded view for Fig. 4.**

(A) Hierarchical clustering of Pearson correlation coefficients between $GR_{inf}$ values for 9 drugs (rows) and 15 circadian clock and growth parameters (columns, averaged *Bmal1-Per2* data). Shown are statistically significant correlation values, where *, and **, indicate *p*-values 0.05, and 0.01, respectively. exact *p*-values: 5-FU-Amplitude = $2.2 \times 10^{-2}$; Alisertib-Lag = $4.1 \times 10^{-2}$; Alisertib-Amplitude = $1.5 \times 10^{-2}$; Alpelisib-Ridge = $1 \times 10^{-2}$; Alpelisib-Rhythmicity = $6.6 \times 10^{-3}$; Alpelisib-Circadian = $1 \times 10^{-2}$; Alpelisib-Infradian = $1.9 \times 10^{-2}$; Alpelisib-Phase diff. var = $2.7 \times 10^{-3}$; Alpelisib-Ultradian = $4.6 \times 10^{-2}$; Olaparib-Growth = $2.7 \times 10^{-2}$; Cisplatin-Rhythmicity = $1 \times 10^{-2}$; Cisplatin-Circadian = $3.2 \times 10^{-2}$; Cisplatin-Infradian = $4.4 \times 10^{-2}$; Cisplatin-Period var. = $3.1 \times 10^{-2}$. $n = 5$ cell lines per drug, except for cisplatin, where $n = 15$ cell lines. (B) Ranking of absolute correlation values between cellular $GR_{inf}$ values and circadian clock/growth parameters, accumulated by individual drugs ($n = 15$ parameters). Bottom and top edges of the boxes represent the 25th and 75th percentiles, respectively. Extending whiskers represent data points within 1.5 times the interquartile range from lower and upper quartile. Red horizontal lines denote median values, red crosses mark outliers. (C) Ranking of absolute correlation values between cellular $GR_{inf}$ values and circadian clock/growth parameters, accumulated by individual parameter ($n = 9$ drugs). See (B) for definition of boxes. (D) See (A), but shown for $GR_{50}$ values. *p*-values: Alpelisib-Amplitude var. = $3.6 \times 10^{-2}$; Cisplatin-Phase diff = $4.8 \times 10^{-2}$; Adavosertib-Period var. = $3.2 \times 10^{-2}$; Paclitaxel-Amplitude = $1.8 \times 10^{-2}$; Doxorubicin-Amplitude = $4.4 \times 10^{-2}$; 5-FU-Lag = $2.5 \times 10^{-2}$; 5-FU=Phase diff. var. = $3.3 \times 10^{-2}$. (E) See (B), but shown for $GR_{50}$ values. (F) See (C), but shown for $GR_{50}$ values. (G) See (A), but shown for Hill coefficient values. *p*-values: Cisplatin-Lag = $2.7 \times 10^{-2}$; Cisplatin-Infradian = $2.5 \times 10^{-2}$; Paclitaxel-Lag = $3 \times 10^{-2}$; Paclitaxel-Phase diff. var. = $3.7 \times 10^{-2}$; Doxorubicin-Amplitude var. = $4.5 \times 10^{-2}$; Doxorubicin-Amplitude decay = $1.2 \times 10^{-2}$; 5-FU-Period = $1.3 \times 10^{-2}$; Torin2-Period = $2.4 \times 10^{-2}$; Adavosertib-Period var. = $3.4 \times 10^{-2}$; (H) See (B), but shown for Hill coefficient values. (I) See (C), but shown for Hill coefficient values.

**Own drug sensitivity dataset**

Cisplatin | DNA damage response inducer

**A**

GR_inf | Circadian gene expression

**B**

GR_50 | Circadian parameters

**C**

GR_50 | Circadian gene expression

**D**

Hill coefficient | Circadian parameters

**E**

Hill coefficient | Circadian gene expression

**F**

**Public CCLE dataset**

Circadian parameters | IC_50

◀ **Figure EV4. Expanded view for Fig. 4.**

(A) Linear discriminant analysis (LDA) on median-binarized cisplatin $GR_{inf}$ values using circadian gene expression data as input. Cell models shown in with $GR_{inf}$ values below or above the median are colored in blue-green and orange, respectively. The right bar plots rank the individual contribution of each input parameter to the obtained discriminative information. (B) See (A), but shown for $GR_{50}$ values and *Bmal1-Per2* oscillation and growth parameters as input. (C) See (A), but shown for $GR_{50}$ values. (D) See (B), but shown for Hill coefficient values and *Bmal1-Per2* oscillation and growth parameters as input. (E) See (A), but shown for Hill coefficient values. (F) LDA profiles of different drugs, exemplifying cell model distributions along LD1 for varying chronosensitivity indices, sorted from highest index (left panel) to lowest (right panel).

