## [Peer Review File · Molecular Systems Biology]

Circadian clock features define novel subtypes among breast cancer cells and shape drug sensitivity

Carolin Ector, Jeff Didier, Sébastien De Landtsheer, Malthe Nordentoft, Christoph Schmal, Ulrich Keilholz, Hanspeter Herzel, Achim Kramer, Thomas Sauter, and Adrián Granada

Corresponding author(s): Adrián Granada (adrian.granada@charite.de)

Review Timeline:

Submission Date:	16th Aug 24
Editorial Decision:	9th Oct 24
Revision Received:	15th Jan 25
Editorial Decision:	10th Feb 25
Revision Received:	11th Feb 25
Accepted:	12th Feb 25

Editor: Jingyi Hou

Transaction Report:

9th Oct 2024

Manuscript Number: MSB-2024-12581

Title: Circadian clock features define novel breast cancer subtypes and shape drug sensitivity

Author: Carolin Ector

Jeff Didier

Sébastien De Landtsheer

Malthe Nordentoft

Christoph Schmal

Ulrich Keilholz

Hanspeter Herzel

Achim Kramer

Thomas Sauter

Adrián Granada

Dear Dr Granada,

Thank you for submitting your work to Molecular Systems Biology. I would like to apologise for the slow process, which was due to the late arrival of reviewers' reports. We have now heard back from two of the three reviewers who agreed to evaluate your manuscript. Unfortunately, after a series of reminders, we did not manage to obtain a report from Reviewer #1. In the interest of time, and since the recommendations of the other two reviewers are quite similar, I prefer to make a decision now rather than further delaying the process. If we receive the comments from Reviewer #1, we will send them to you, and you can address the issues raised by Reviewer #1 together with those raised by the other two reviewers. You will see from the comments below that Reviewers #2 and #3 find the manuscript to be of interest. They raise, however, several points, which should be convincingly addressed in a revision of this work.

I think that the reviewers' recommendations are rather clear and there is no need for me to reiterate their comments. In particular, Reviewer #2 suggested performing cross-validation to confirm the results, which should be carefully addressed. All other issues raised by the reviewers need to be satisfactorily addressed. Please contact me in case you would like to discuss in further detail any of the issues raised. As you may already know, our editorial policy allows in principle a single round of major revision, and it is therefore essential to provide responses to the reviewers' comments that are as complete as possible.

On a more editorial level, we would ask you to address the following issues:

- Please provide a .docx formatted version of the manuscript text (including legends for main figures, EV figures and tables). Please make sure that the changes are highlighted to be clearly visible.
- Please provide individual production quality figure files as .eps, .tif, .jpg (one file per figure).
- Please provide a .docx formatted letter INCLUDING the reviewers' reports and your detailed point-by-point responses to their comments. As part of the EMBO Press transparent editorial process, the point-by-point response is part of the Review Process File (RPF), which will be published alongside your paper.
- Please note that all corresponding authors are required to supply an ORCID ID for their name upon submission of a revised manuscript.
- We replaced Supplementary Information with Expanded View (EV) Figures and Tables that are collapsible/expandable online (see examples in <http://msb.embopress.org/content/11/6/812>). A maximum of 5 EV Figures can be typeset. EV Figures should be cited as 'Figure EV1, Figure EV2' etc... in the text and their respective legends should be included in the main text after the legends of regular figures.

Additional Tables/Datasets should be labeled and referred to as Table EV1, Dataset EV1, etc. Legends have to be provided in a separate tab in case of .xls files. Alternatively, the legend can be supplied as a separate text file (README) and zipped together with the Table/Dataset file.

For the figures and tables that you do NOT wish to display as Expanded View figures, they should be bundled together with their legends in a single PDF file called *Appendix*, which should start with a short Table of Content. Each legend should be below the corresponding Figure/Table in the Appendix. Appendix figures and tables should be referred to in the main text as: "Appendix Figure S1, Appendix Figure S2, Appendix Table S1" etc. See detailed instructions regarding expanded view here: <https://www.embopress.org/page/journal/17444292/authorguide#expandedview>.

- Before submitting your revision, primary datasets (and computer code, where appropriate) produced in this study need to be

deposited in an appropriate public database (see <http://msb.embopress.org/authorguide> - data availability <https://www.embopress.org/page/journal/17444292/authorguide#dataavailability>).

The accession numbers and database should be listed in a formal "Data Availability" section (placed after Materials & Method) that follows the model below (see also <https://www.embopress.org/page/journal/17444292/authorguide#dataavailability>). Please note that the Data Availability Section is restricted to new primary data that are part of this study.

Data availability

- RNA-Seq data: Gene Expression Omnibus GSE46843 (<https://www.ncbi.nlm.nih.gov/geo/query/acc.cgi?acc=GSE46843>)

- [data type]: [name of the resource] [accession number/identifier/doi] ([URL or identifiers.org/DATABASE:ACCESSION])

- At EMBO Press we ask authors to provide source data for the main figures. Our source data coordinator will contact you to discuss which figure panels we would need source data for and will also provide you with helpful tips on how to upload and organize the files.

- Our journal encourages inclusion of *data citations in the reference list* to directly cite datasets that were re-used and obtained from public databases. Data citations in the article text are distinct from normal bibliographical citations and should directly link to the database records from which the data can be accessed. In the main text, data citations are formatted as follows: "Data ref: Smith et al, 2001". In the Reference list, data citations must be labeled with "[DATASET]". A data reference must provide the database name, accession number/identifiers and a resolvable link to the landing page from which the data can be accessed at the end of the reference. Further instructions are available at .

- We updated our journal's competing interests policy in January 2022 and request authors to consider both actual and perceived competing interests. Please review the policy <https://www.embopress.org/competing-interests> and update your competing interests if necessary.

Please use the heading "Disclosure statement and competing interests".

- All Materials and Methods need to be described in the main text using our 'Structured Methods' format. According to this format, the Methods section includes a Reagents and Tools Table (listing key reagents, experimental models, software and relevant equipment and including their sources and relevant identifiers) followed by a Methods and Protocols section describing the methods, ideally using a step-by-step protocol format. The aim is to facilitate adoption of the methodologies across labs. Please download and fill our Reagents and Tools Table template (.docx), which you can find in our author guidelines: <https://www.embopress.org/page/journal/17444292/authorguide#structuredmethods>.

- Regarding data quantification:

Please ensure to specify the name of the statistical test used to generate error bars and P values, the number (n) of independent experiments (please specify technical or biological replicates) underlying each data point and the test used to calculate p-values in each figure legend. Discussion of statistical methodology can be reported in the materials and methods section, but figure legends should contain a basic description of n, P and the test applied.

Graphs must include a description of the bars and the error bars (s.d., s.e.m.).

- Please provide a "standfirst text" summarizing the study in one or two sentences (approximately 250 characters, including space), three to four "bullet points" highlighting the main findings and a "synopsis image" (550px width and 400-600 px height, PNG format) to highlight the paper on our homepage.

Here are a couple of examples:

<https://www.embopress.org/doi/10.15252/msb.20199356>

<https://www.embopress.org/doi/10.15252/msb.20209475>

<https://www.embopress.org/doi/10.15252/msb.209495>

When you resubmit your manuscript, please download our CHECKLIST (<https://www.embopress.org/pb-assets/embosite/EMBO%20Press%20Author%20Checklist-1642513524327.xlsx>) and include the completed form in your submission.

Please note that the Author Checklist will be published alongside the paper as part of the transparent process (<https://www.embopress.org/page/journal/17444292/authorguide#transparentprocess>).

If you feel you can satisfactorily deal with these points and those listed by the referees, you may wish to submit a revised version of your manuscript. Please attach a covering letter giving details of the way in which you have handled each of the points raised

by the referees. A revised manuscript will be once again subject to review and you probably understand that we can give you no guarantee at this stage that the eventual outcome will be favorable.

I look forward to receiving your revised manuscript soon.

Yours sincerely,

Jingyi Hou, PhD
Scientific Editor
Molecular Systems Biology

We realize that it is difficult to revise to a specific deadline. In the interest of protecting the conceptual advance provided by the work, we recommend a revision within 3 months (7th Jan 2025). Please discuss the revision progress ahead of this time with the editor if you require more time to complete the revisions. Use the link below to submit your revision:

IMPORTANT: When you send your revision, we will require the following items:

1. the manuscript text in LaTeX, RTF or MS Word format
2. a letter with a detailed description of the changes made in response to the referees. Please specify clearly the exact places in the text (pages and paragraphs) where each change has been made in response to each specific comment given
3. three to four 'bullet points' highlighting the main findings of your study
4. a short 'blurb' text summarizing in two sentences the study (max. 250 characters)
5. a 'thumbnail image' (550px width and max 400px height, Illustrator, PowerPoint or jpeg format), which can be used as 'visual title' for the synopsis section of your paper.

6. Please include an author contributions statement after the Acknowledgements section (see

<https://www.embopress.org/page/journal/17444292/authorguide>)

7. Please complete the CHECKLIST available at (<https://bit.ly/EMBOPressAuthorChecklist>).

Please note that the Author Checklist will be published alongside the paper as part of the transparent process

(<https://www.embopress.org/page/journal/17444292/authorguide#transparentprocess>).

See also figure legend guidelines: <https://www.embopress.org/page/journal/17444292/authorguide#figureformat>

9. Please note that corresponding authors are required to supply an ORCID ID for their name upon submission of a revised manuscript (EMBO Press signed a joint statement to encourage ORCID adoption).

(<https://www.embopress.org/page/journal/17444292/authorguide#editorialprocess>)

Currently, our records indicate that the ORCID for your account is 0000-0003-0537-9091.

Link Not Available

11. Include a Reagents and Tools Table as part of the Methods section, which can be downloaded from our author guidelines (<https://www.embopress.org/page/journal/17444292/authorguide#structuredmethods>)

*** PLEASE NOTE *** As part of the EMBO Press transparent editorial process initiative (see our Editorial at <https://dx.doi.org/10.1038/msb.2010.72>), Molecular Systems Biology publishes online a Review Process File with each accepted manuscripts. This file will be published in conjunction with your paper and will include the anonymous referee reports, your point-by-point response and all pertinent correspondence relating to the manuscript. If you do NOT want this File to be published, please inform the editorial office at msb@embo.org within 14 days upon receipt of the present letter.

Reviewer #2:

Summary

This manuscript follows up on the author's recent publication Ector et al. (<https://www.nature.com/articles/s41467-024-51611-3>) where they develop most of the methods / analysis strategies used in herein. In the manuscript, they focus on breast cancer cell lines (also including human osteosarcoma U2OS) and aim to define novel circadian clock-based phenotypes in these cell lines and to uncover if and how circadian features correlate with drug sensitivity and tumor growth.

They employ data analysis methods to deep-phenotype the circadian clock and cross correlate this with cellular growth dynamics, drug responses, and genomic and transcriptomic profiling of core circadian clock genes from literature databases. They then attempt to integrate the estimated cellular parameters to identify novel circadian based phenotype clusters in the analysed breast cancer cell lines. Finally, they study the effect of circadian clock on drug sensitivity, identifying some potential targets for further investigation.

Overall, we think this is a timely and important idea using a very rich and multidimensional dataset. The scope of the work is most interesting and following not only from the authors previous papers, but also other groups interested in this topic (e.g., Lee et al. DOI: 10.1126/sciadv.abd2645) but significantly extend this in a novel direction. The results and interpretation are consistent and make sense within the frame of the work, but their utility for further translational studies and ultimately clinical decision-making remains to be established.

Major comments

- It is unclear to what extent the results might be transferrable to the clinical situation. The studied circadian parameters have been shown to be liable to culturing conditions (e.g., from Meng lab about matrix, the Kramer group on density of cells) and the relationship to in vivo patient tumors is still to be established. Thus, using "define novel breast cancer subtypes" in the title seems unwarranted and it should be pointed out that these results come from a modest number of cell line models. Otherwise, the authors would need to find a link to at least patient derived models like the recent paper from Anafi/Meng groups. (also see line 48 in abstract "breast cancer").

- The logic of lines 223-238 is unclear. By definition PCA identifies the axis of the most variation, which are then clustered in some way. The authors then state that they don't directly translate into circadian phenotype, but take the top identifying feature from PCA for each cluster, isolate them and re-cluster using k-means to claim that they successfully clustered cells. What does successfully mean?

One would normally expect to end up with some clusters especially with this many data points (even if the data was random). Can the authors please clarify 'successfully classified cells into four circadian phenotypes' and given this is quite an important claim, substantiate it further. For example, using UMAP to cluster would have been different although UMAP on this many data points would also tend to produce spurious clusters (especially with n neighbors equal to 3).

- The authors use min-max scaling of the circadian parameters for PCA but this lacks unit variance. Standard is zero mean and unit variance. Was there any reason for min-max and are the results affected if using a different transformation?

- The authors calculated Pearson correlation coefficients across different circadian parameters and growth rates and 'averaged them'. How was the averaging done? Using Fisher's z-transformation? This should be better described.

- They use LDA 'to discern the cumulative effect of circadian features on categorizing cell models into groups of high or low sensitivity'. If we understand correctly, sensitivity is available as a continuous metric, so why not use ridge/lasso type regression not to lose this information? Or logistic regression if they want to classify. As logistic regression this should be more robust in terms of underlying assumption than the LDA. Might LR be more appropriate to study the underlying structure of the prediction? This really seems to be what they are interested in.

- A chronosensitivity index from a linear model would be more straightforward in terms of goodness-of-fit / anova type tests. Why wasn't this used?

- It would be important to have some cross-validation to confirm the result (also see general point about transfer to clinical situation above). If this is a robust metric, as they claim, this should be attempted?

- Another point about the in vitro to in vivo comparison might be the mouse and baboon temporal atlases (Zheng et al, Mure et al) and the more recent work in human tissue rhythms (Talamanca et al) all suggesting that the phasing in vivo is similar across tissue. In Figure 2E/F (should be other way round?) there is quite a difference in phase angles even in the rhythmic cell lines that is beyond the expected in vivo situation (?). This should be discussed further.

- Lines 395 to 400, the authors claim that the mutation burden correlates with clock function but figure 3E shows that dysfunctional clocks are on par with functional with weak and unstable higher but is their explanation for this? This is not mentioned in the main body of the paper so the discussion should be toned down in this point.

- Lines 268 to 275: Using transcriptomic profiles from unsynchronised cell lines to claim misregulation of one or the other core clock gene between cell lines is associated with classical BC subtype but not their clock-based categories. Although there are many examples in the literature of doing this (admittedly), especially given the claim that some of these do have functional clocks while others have not does not, it should be mentioned that this might be most misleading as it is well known that overall transcriptional activities (see PCNA) are different in different breast cancers and therefore different levels of expression might be related to different cell lines overall transcriptional activities rather than carry any clock related information.

- It seems that growth rate might have the potential to influence the bioluminescence reporter recordings, even without invoking the coupling between cells, very fast cell cycling like in the MDAMB231 cells might make the clock appear "worse" than single cell recordings would suggest. This might also influence the drug sensitivity (also in comparison to Lee et al). We don't suggest replicating the findings with single cell recording. However, this should be discussed explicitly.

Minor comments

- The authors write: "Through the introduction of the "chronosensitivity index", we provide a robust metric for quantifying the

circadian clock's impact on differentiating between high and low drug sensitivity groups". The ratio of between to within cluster distances is one of the many standard metrics to assess the quality of clustering, if not being the most advanced. Why they wouldn't just call it that?

- Line 671: They did not mean to cite Balsalobre's paper here.

Reviewer #3:

The manuscript by Dr. Ector and colleagues, titled "Circadian Clock Features Define Novel Breast Cancer Types and Shape Drug Sensitivity," focuses on the application of circadian oscillators' properties to redefine subtypes of breast cancer, as suggested by the title. The authors set a highly ambitious goal to redefine classical breast cancer phenotypes by combining in-depth phenotyping of cellular circadian oscillators with analyses of cell growth, genomic properties, and drug responses. To achieve this, the authors applied their own recently published novel methodological approach (from this year's Nature Communications publication by the same group), which they extend in terms of analysis, to 19 immortalized cell lines (both malignant and benign).

Multiresolution analysis was employed to assess different time components, followed by an assessment of the signal-to-noise ratio for the circadian component. This parameter was further validated using the U2OS cell line, which lacks either CRY proteins alone or both Cry1/Cry2. These state-of-the-art analyses, designed to define circadian clock-based cancer cell phenotypes, were completed by assessing phase coherence between two anti-phasic circadian reporters (Bmal1-luc and Per2-luc) introduced into each cell line, to further define the coupling strength for each cancer subtype. These analyses were then combined with mutational assessments.

The proposed clock phenotyping was further correlated with cell growth rate. Strikingly, the circadian phenotyping proposed here did not fully overlap with classical subtyping, opening new possibilities for precision medicine based on optimal therapeutic sensitivity as defined by circadian phenotyping. To further explore this novel classification, the authors screened a wide spectrum of classical anticancer agents to correlate drug responsiveness with clock parameters. The correlations varied widely depending on the nature of the compounds, with distinct contributions from each of the core clock genes, which were analyzed individually, identifying Rora and Cry2 as the most significant contributors.

The circadian phenotyping proposed here holds outstanding promise, not only for understanding the roles of core clock machinery in breast cancer progression but also for developing personalized therapeutic approaches (including chronotherapy) better tailored to the malignancy characteristics. In summary, this study represents a valuable advancement from both basic and preclinical perspectives; it is well-designed and clearly presented.

Minor comments

1. While immortalized cells are an indispensable model for basic studies and enable high-throughput analyses like those conducted by the authors, using primary cells may represent an important next step towards the preclinical stage. Although these approaches are beyond the scope of the current study, future application of the proposed methodology to primary human cells derived from malignant and non-malignant tissue biopsies, and the feasibility of such an approach, could be discussed.
2. A recent publication by Manella and colleagues (PMID: 34625543) presents a novel method for rapidly establishing clock-resetting properties (CIRCA-Scope). Do the authors consider this method as complementary to the analyses presented here (e.g., clock resetting upon therapeutic administration in a time-of-day dependent manner)? If so, this point could be discussed.
3. One of the major novelties of this work is the method for in-depth circadian phenotyping combined with additional parameters. It would be of great value to the field if the authors made the developed methodology accessible to the reader in a straightforward manner, upon publication of the manuscript.

We sincerely thank the reviewers for taking their time to evaluate our manuscript and for offering valuable feedback. In this response, we have carefully addressed each of the concerns raised, including new analyses in the revised manuscript. For ease of review, we are providing a highlighted version of the manuscript to indicate the changes. Modifications made independently of the reviewers' suggestions are detailed at the end of this letter.

****Reviewer #2****

Summary

This manuscript follows up on the author's recent publication Ector et al. (<https://www.nature.com/articles/s41467-024-51611-3>) where they develop most of the methods / analysis strategies used in herein. In the manuscript, they focus on breast cancer cell lines (also including human osteosarcoma U2OS) and aim to define novel circadian clock-based phenotypes in these cell lines and to uncover if and how circadian features correlate with drug sensitivity and tumor growth. They employ data analysis methods to deep-phenotype the circadian clock and cross correlate this with cellular growth dynamics, drug responses, and genomic and transcriptomic profiling of core circadian clock genes from literature databases. They then attempt to integrate the estimated cellular parameters to identify novel circadian based phenotype clusters in the analysed breast cancer cell lines. Finally, they study the effect of circadian clock on drug sensitivity, identifying some potential targets for further investigation. Overall, we think this is a timely and important idea using a very rich and multidimensional dataset. The scope of the work is most interesting and following not only from the authors previous papers, but also other groups interested in this topic (e.g., Lee et al. DOI: 10.1126/sciadv.abd2645) but significantly extend this in a novel direction. The results and interpretation are consistent and make sense within the frame of the work, but their utility for further translational studies and ultimately clinical decision-making remains to be established.

We thank the reviewer for their evaluation of our manuscript and for recognizing the relevance of this topic. In response to the reviewer's comments, we have expanded on our analysis approaches, added further explanations to our methodological approaches, and made the limitations of our study clearer. We believe that these changes have significantly enhanced the overall readability and presentation of our work. Below, we address each of the reviewer's major and minor points individually.

Major comments

1. It is unclear to what extent the results might be transferrable to the clinical situation. The studied circadian parameters have been shown to be liable to culturing

conditions (e.g., from Meng lab about matrix, the Kramer group on density of cells) and the relationship to in vivo patient tumors is still to be established. Thus, using "define novel breast cancer subtypes" in the title seems unwarranted and it should be pointed out that these results come from a modest number of cell line models. Otherwise, the authors would need to find a link to at least patient derived models like the recent paper from Anafi/Meng groups. (also see line 48 in abstract "breast cancer").

We appreciate the reviewer's critical insights regarding the transferability of our results to clinical situations. We agree that the applicability of our findings to in vivo patient tumors has not yet been established, and therefore, we have revised our manuscript title to explicitly refer to cancer cell models. Additionally, we have adjusted the abstract (page 2, line 43-44) to more accurately reflect the scope of our in vitro study.

In addition, we have further contextualized the limitations of our work in the discussion section of our manuscript (see page 15, lines 490-496).

2. The logic of lines 223-238 is unclear. By definition PCA identifies the axis of the most variation, which are then clustered in some way. The authors then state that they don't directly translate into circadian phenotype, but take the top identifying feature from PCA for each cluster, isolate them and re-cluster using k-means to claim that they successfully clustered cells. What does successfully mean? One would normally expect to end up with some clusters especially with this many data points (even if the data was random). Can the authors please clarify 'successfully classified cells into four circadian phenotypes' and given this is quite an important claim, substantiate it further. For example, using UMAP to cluster would have been different although UMAP on this many data points would also tend to produce spurious clusters (especially with n neighbors equal to 3).

We appreciate the reviewer's thorough examination of our approach to clustering cell lines based on their circadian properties and regret the initial lack of clarity in our explanation. We have now refined our assessment and further substantiated the quality of our results.

We acknowledge that our use of the term "successfully" was colloquial and potentially unclear. In the original version of our manuscript, we used the term "successfully classified" to describe our assessment of the biologically meaningful and well-separated cell line clusters identified using the k-means clustering approach. To address potential ambiguity, we have revised the manuscript to remove the term "successfully" and ensure clarity (page 8, line 225).

In addition, we would like to provide further information regarding our clustering strategy and how we evaluated its applicability in defining specific clusters. We used

the elbow method to determine the optimal number of clusters. This method involves plotting the within-cluster sum of squares (WCSS) against a range of cluster numbers (k , from 2 to 7) and identifying the 'elbow' point at which the WCSS begins to decrease less dramatically, indicating a suitable number of clusters beyond which there is diminishing improvement in variance explained. This analysis supports our decision to cluster the cell lines into four distinct circadian phenotypes (see **Response Letter Figure-1** in this review file). This is supported by a WCSS value of 0.79 and a Silhouette score of 0.45, both suggesting a reasonable level of cluster separation, especially given our small dataset of 15 cell lines. This analysis informed our assessment of cluster separability.

Response Letter Figure-1 | Statistical validation of k-means clustering of cell line models by circadian parameters. Elbow plot illustrating the within-cluster sum of squares (WCSS) for varying numbers of clusters (k). The optimal number of clusters ($k = 4$) is indicated with the red dotted line, marking the elbow point where increasing the number of clusters no longer provides significant benefits. Silhouette scores are displayed within the plot's boxes to assess cluster quality, where 1 indicates clear separation, 0 moderate, and -1 poor separation between clusters.

To further analyze our clustering results, we performed a complementary Uniform Manifold Approximation and Projection (UMAP) analysis on the 3 selected circadian features. The UMAP clustering visually confirmed the presence of four distinct circadian phenotypes using 2–4 neighbours (n) (see **Response Letter Figure-2** in this review file), which aligned with our k -means clustering findings.

Response Letter Figure-2 | Validation of the four cell line clusters using UMAP on the selected circadian features. UMAP's mapping the cell line models by the three selected circadian features used for k-means clustering. Different numbers of neighbours (n) were chosen, ranging from 2 to 4, with $n = 2$ and 3 showing the best separability between the identified clock phenotypes (color coded). The classical subtype of each model is illustrated by different markers.

To better clarify the rationale for phenotyping the cell line models based on k-means clustering, we now include the elbow plot as **Appendix Figure S3** and the complementary UMAP analysis in **Figure EV2C** in the revised manuscript and refer to them in lines 233–239 on page 8 of the revised manuscript.

We hope that this explanation makes our approach and clustering strategies more transparent and understandable.

3. The authors use min-max scaling of the circadian parameters for PCA but this lacks unit variance. Standard is zero mean and unit variance. Was there any reason for min-max and are the results affected if using a different transformation?

We regret that the original manuscript did not clarify the rationale behind our choice of normalization strategy. In response to the feedback provided, we expand our discussion on this topic and evaluated additional normalization strategies in the following.

In our original analysis, we employed min-max scaling to normalize the circadian parameters because it preserves the original distribution of the data while ensuring that all features are constrained between 0 and 1, accommodating their differing units and ranges. This method is crucial for maintaining the integrity of the original data distribution in our study.

To address the concern regarding the lack of unit variance with min-max scaling, we performed a comparative analysis using both standard zero-mean and unit variance scaling (equation 2) alongside our initial min-max approach (equation 1). This new analysis revealed that while standard scaling addresses unit differences effectively, it is susceptible to distortion by outliers, which could skew the PCA results. Conversely, min-max scaling not only ensures that values are uniformly distributed between 0 and

1 but also retains the original distribution of the data. These comparative results have been illustrated in **Response Letter Figure-3**.

$$\text{Min-max scaling} = \frac{X_i - X_{\text{minimum}}}{X_{\text{maximum}} - X_{\text{minimum}}} \quad (\text{eq. 1})$$

$$\text{Standard scaling} = \frac{X_i - X_{\text{mean}}}{X_{\text{standard deviation}}} \quad (\text{eq. 2})$$

Response Letter Figure-3 | Data distribution for different normalization techniques. Raw values exhibit varying units and ranges (left panel). Standard-scaled values normalize for the varying units, but outliers change the original distribution of the data, skewing differences between parameters (middle panel). Min-max-scaling preserves the original distribution and distribute values between 0 and 1.

To assess how the different transformation methods such as standard scaling affect the PCA results, we performed complementary PCA using the standard-scaled values and found that the overall insights remained consistent with those obtained using min-max scaling (see **Response Letter Figure-4** of this review letter and compare to **Figure 3B** of the main manuscript). The primary difference was minor variations in the weight each feature contributed to the clustering, but these did not significantly affect the identification of the circadian phenotypes.

Response Letter Figure-4 | Principal component analysis on standard-scaled circadian and growth features. (A) PCA biplot of the indicated circadian and growth parameters, displaying the distribution of 15 cell line models across the first two principal component scores. Cell models are depicted by different markers indicating their classical subtype. Color-coded clusters are outlined around closely positioned cell models, based on visual inspection. The variance (PC loadings) explained by each component is expressed in percentage. CV=coefficient of variation. (B) Ranking of absolute PC loadings for each circadian and growth parameter, corresponding to the PCA biplot shown in A. Parameters are sorted in descending order based on their absolute contribution in the first principal component. Negative values are shaded. Overall, these results are in alignment with results obtained from transforming min-max-scaled values.

We now have added an explanation for our choice of min-max scaling in the revised Methods section on page 23, lines 700–703.

- The authors calculated Pearson correlation coefficients across different circadian parameters and growth rates and 'averaged them'. How was the averaging done? Using Fisher's z-transformation? This should be better described.

We apologize for not clearly describing this method in the original version of our manuscript. In the Methods section, when we mentioned that "*The relationship across different circadian parameters and growth rates were computed and interpreted using the Pearson correlation coefficient and corresponding p-values on combined Bmal1-Per2 data, averaged across breast cancer cell lines,*" we intended to explain that we first calculated the mean of the Bmal1-Per2 circadian parameters across the breast cancer cell lines. We then calculated the Pearson correlation coefficients among these averaged circadian and growth parameters.

We have clarified more clearly this process in the revised text, now found on page 23 in lines 693–697. Thank you for bringing this to our attention.

5. They use LDA 'to discern the cumulative effect of circadian features on categorizing cell models into groups of high or low sensitivity'. If we understand correctly, sensitivity is available as a continuous metric, so why not use ridge/lasso type regression not to lose this information? Or logistic regression if they want to classify. As logistic regression this should be more robust in terms of underlying assumption than the LDA. Might LR be more appropriate to study the underlying structure of the prediction? This really seems to be what they are interested in.

We thank the reviewer for their suggestion, which prompted us to implement and test the additional analyses they recommended, thereby providing us with new insights. Before discussing these new analyses, it's important to explain the rationale behind our original choice of using LDA, which was primarily dictated by the sample size constraints in our study. LDA is particularly appropriate for datasets with smaller sample sizes, where it effectively reduces dimensionality while maintaining class separability^{1, 2}. However, we agree with the reviewer that Logistic Regression (LR) constitutes an attractive alternative for classification without forcing assumptions about the data distribution. Furthermore, classifying cell models based on multiple regression models that preserve the continuity in drug sensitivity values is intriguing.

In response to the reviewer's suggestions, we now conducted additional ridge, lasso, and logistic regression analyses to evaluate the dependency of drug sensitivity metrics on circadian and growth features. For this, we expanded our original set of three cisplatin sensitivity parameters by the area over the curve (GR_{AOC}) and the concentration at half maximal effect (GEC_{50}). To systematically assess and compare the accuracy of each approach, we calculated the accuracy based on two distinct approaches depending on the regression or classification model:

- (i) For the new ridge and lasso regression, we calculated the difference between actual and predicted sensitivity values and related that value to highest value among the true values for each sensitivity metric. Based on that relative difference, we classified cell lines as correctly predicted if the difference was less than 10%.
- (ii) For the original LDA and the new complementary LR classification models, we performed leave-one-out-cross-validation (LOOCV) and calculated the accuracy based on correctly classified cell line models.

The accuracy results of these new analyses are compiled in **Response Letter Figure-5**. Overall, the mean accuracies across sensitivity values for all four analysis methods are moderate to low, likely due to the limited sample size and varying dependencies on the cell line model and drug sensitivity parameter (**Response Letter Figure-5**). However, the ridge and lasso regression approaches are clearly outperformed by LDA and LR regression, with mean accuracies across all drug sensitivity parameters differing by up to ~2-fold. LDA achieved a mean accuracy of 0.42 while LR reached

0.51, making LR the most accurate model for this dataset. To address similar concerns future readers might have about our choice for LDA, we now include the below **Response Letter Figure-5** in the new Appendix Figure S4A-D.

Response Letter Figure-5 | Accuracy of complementary analysis methods for predicting circadian-dependent drug sensitivity. Comparison of the accuracies of the original LDA with additional methods, namely lasso regression, ridge regression, and **Error! Reference source not found.** regression, in predicting drug sensitivity based on circadian and growth parameters. *Upper panel:* Predictive performance using lasso and ridge regression. The differences (d) between predicted (p) and true (t) sensitivity values are color-coded as follows: shades of green indicate discrepancies below 10%, with darker green indicating smaller differences; yellow to red tones indicate discrepancies above 10%, with dark red representing the highest discrepancies. Additionally, R^2 (r^2) from the regression fits between actual and predicted sensitivity values are displayed. *Lower panel:* Leave-one-out-cross-validation (LOOCV) analysis comparing the original LDA analysis and the new logistic regression. Green boxes indicate correct classification of the respective cell line model, while red signifies misclassified cases. Grey boxes = no data. Accuracy in white boxes. The cell lines are sorted as above. The mean accuracy among all drug sensitivity metrics is stated below the regression methods.

To assess which classification metric, LDA or LR, is better suited for our entire dataset, we applied both methods to the publicly available CCLE drug sensitivity dataset, for which we also included two additional sensitivity metrics, namely the half-effective drug concentration (EC_{50}) and the area above the response curve (ActArea). In contrast to the cisplatin dataset using circadian parameters as input, LDA consistently resulted in higher overall accuracy than LR. This was observed for both, circadian and growth parameters as input or circadian gene expression values (**Response Letter Table-1**). Based on these new insights, we decided to continue to

use LDA for our exploration of circadian dependencies in drug sensitivity. However, to facilitate individualized assessments of accuracy by sensitivity metric and cell line model, we now include the LOOCV-based accuracy results for all dataset-input-parameter-combinations in the new Appendix Figure S4 and S5 of the revised manuscript. We thank the reviewer for this suggestion, which has enriched our study. The new analysis prompted by their feedback is thoroughly described in lines 337–350 on page 11 and on page 12 (lines 364–370 and 376–382) of the revised manuscript.

Drug sensitivity dataset	Sensitivity parameter	Model	Mean accuracy by input	
			Circadian parameters	CCLL gene expression
CCLE (various drugs; see 4H, I of the manuscript)	IC ₅₀ (originally shown in manuscript)	LDA	0.33	0.47
		Logistic regression	0.29	0.38
	EC ₅₀ (new)	LDA	0.32	0.38
		Logistic regression	0.14	0.21
	ActArea (new)	LDA	0.28	0.24
		Logistic regression	0.18	0.20
Own dataset (cisplatin)	Combined (GR _{inf} , GR ₅₀ , Hill [original]; GR _{AOC} , GEC ₅₀ [new]).	LDA	0.42	0.51
		Logistic regression	0.51	0.46

Response Letter Table-1 | Mean accuracies of LDA and Logistic Regression (LR) in classifying drug sensitivity groups. Overview of mean accuracy values of LDA and LR for classifying cell models into drug sensitivity groups based on circadian parameters. For the CCLE dataset, the mean accuracy is determined by averaging the accuracies across different drugs. In our cisplatin sensitivity dataset, the mean accuracy is calculated by combining various sensitivity metrics. Underlined values highlight the higher accuracy between LDA and LR for each dataset.

6. A chronosensitivity index from a linear model would be more straightforward in terms of goodness-of-fit / anova type tests. Why wasn't this used?

This idea is certainly interesting; unfortunately, due to the current sample size of our dataset, regression models such as linear or ANOVA-based approaches proved too imprecise for reliable analysis. The small number of cell lines led to insufficient statistical power, resulting in high variability and reduced predictive accuracy (see major comment 5 and **Response Letter Figure-5**). Consequently, we opted to use the LDA classification method, complemented by LOOCV of both LDA and additional logistic regression in the supplements (Appendix Figure S4 and S5). These methods aim to provide more stable and meaningful results under the constraints of our dataset. With an expanded sample size in future studies, we would be very keen to explore this approach further.

7. It would be important to have some cross-validation to confirm the result (also see general point about transfer to clinical situation above). If this is a robust metric, as they claim, this should be attempted?

This is a very important point. To address this issue, we now employed leave-one-out-cross-validation (LOOCV) for both our original linear discriminant analysis (LDA) and the complementary logistic regression (LR) models, as described in major comment 5. We considered LOOCV to be particularly suitable for our study due to the limited sample size, as it maximizes the use of available data by iteratively training the model on all but one data point and testing on the excluded point. This approach provides a nearly unbiased estimate of the model's performance.

We now comprehensively show the accuracy results from the LOOCV in Appendix Figure S4 and S5 of the revised manuscript and mean accuracies are summarized in **Response Letter Table-1** of this response file.

In summary, our new analysis indicates that both, LDA and LR achieved overall moderate and largely similar accuracies, with LDA generally outperforming LR across most scenarios. The accuracy seems to depend strongly on the (i) drug, (ii) cell line model, (iii) input dataset (circadian parameters or gene expression), and (iv) the drug sensitivity value. Strikingly, the drug with the highest ranking chronosensitivity index using circadian parameters, 17-AAG, showed the highest accuracy in LOOCV analysis on the LDA results (71%), despite the low sample number of 7 cell lines. Interestingly, Lee et al. (2021) identified the HSP90 inhibitor 17-AAG as one of the highest ranked drugs for time-of-day variability using an experimentally-based approach³. On the other hand, the lowest ranking "chronosensitivity drug", topotecan, exhibited the poorest accuracy (14%). The complementary LOOCV of the logistic regression approach further predicted high accuracies of 0.71 for the next 3 highest ranking drugs, namely erlotinib, nilotinib, and AZD0530. Similarly, the top 3 scoring drugs in the chronosensitivity index based on circadian gene expression data were also the most accurately performing drugs in the LOOCV on the LDA (compare Figure 4H with Appendix Figure S5A, B and Figure 4I with Appendix Figure S5C, D).

Despite these results from the LOOCV for these highlighted drugs, in many cases we found low accuracies for other drugs and the overall small sample size based on 2D cell lines warrant for further validation with larger and more diverse datasets. Especially for the translatability to clinical situations, further studies in more complex tumor models such as patient-derived organoids or patient-based clinical studies are needed.

In summary, we are confident that the added cross-validation analysis and the exploration of different classification approaches added metrics of confidence to our results and approach, while also exposing needed areas for further validation. We

refer to these new findings on page 12 in lines 364–370 and 376–382, and sincerely thank the reviewer for guiding us to this advancement of our findings.

8. Another point about the in vitro to in vivo comparison might be the mouse and baboon temporal atlases (Zheng et al, Mure et al) and the more recent work in human tissue rhythms (Talamanca et al) all suggesting that the phasing in vivo is similar across tissue. In Figure 2E/F (should be other way round?) there is quite a difference in phase angles even in the rhythmic cell lines that is beyond the expected in vivo situation (?). This should be discussed further.

We thank the reviewer for highlighting the importance of providing appropriate context for our observed phase differences. Studies like the mentioned ones by Talamanca et al.⁴, Zhang et al.⁵ (we believe Zhang is meant by the reviewer, not Zheng), and Mure et al.,⁶ provide indeed valuable insights into conserved circadian phase relationships across tissues in vivo. The differences in our findings and these studies are potentially based on the distinct experimental contexts, as highlighted by the reviewer's concern about in vitro versus in vivo comparisons.

All of these mentioned studies use biological tissues to explore systemic and tissue-specific circadian rhythms. Talamanca et al. examined human tissues⁴, Zhang et al. studied transcriptional rhythms in mouse organs⁵, and Mure et al. focused on baboon tissues⁶. These approaches are able to capture the influence of systemic cues and the orchestration of circadian rhythms across the organism. In contrast, our study focused on phase differences between *Bmal1* and *Per2* signals in isolated cell lines. Without the systemic signals present in vivo, our in vitro models naturally show a broader range of phase differences, especially within breast cancer subtypes. Additionally, our in-depth circadian phenotyping approach uncovered significant heterogeneity in the properties of the circadian clock, which could contribute to the observed variability in *Bmal1*-*Per2* phase differences. This suggests that there are subtype-specific differences in circadian behavior within cancer tissues.

Acknowledging the need to better contextualize our description of the observed phase differences, we have now revised the manuscript and have expanded our discussion to include comparisons with the three mentioned studies (page 13, lines 403–415). (*Note: we indeed intended to show the individual subtypes in E first and then the composite in F and arranged the figures like that due to space constraints*).

9. Lines 395 to 400, the authors claim that the mutation burden correlates with clock function but figure 3E shows that dysfunctional clocks are on par with functional with weak and unstable higher but is their explanation for this? This is not mentioned in the main body of the paper so the discussion should be toned down in this point.

We thank the reviewer for highlighting the need for a clearer explanation regarding the relationship between mutation burden and clock functionality as depicted in Figure 3E. In the revised Discussion section (page 14, lines 438–447), we have clarified that while high mutation burdens are associated with weak and unstable clock phenotypes, dysfunctional clocks do not show a significantly higher mutation burden compared to functional clocks. This distinction emphasizes that factors beyond mutation burden, i.e. non-genetic factors such as epigenetic modifications (as recently reviewed by Liu et al.⁷), may contribute to clock dysfunction.

10. Lines 268 to 275: Using transcriptomic profiles from unsynchronised cell lines to claim misregulation of one or the other core clock gene between cell lines is associated with classical BC subtype but not their clock-based categories. Although there are many examples in the literature of doing this (admittedly), especially given the claim that some of these do have functional clocks while others have not does not, it should be mentioned that this might be most misleading as it is well know that overall transcriptional activities (see PCNA) are different in different breast cancers and therefore different levels of expression might be related to different cell lines overall transcriptional activities rather than carry any clock related information.

We appreciate the reviewer's important insight regarding the potential confounding effects of overall transcriptional activity differences across breast cancer subtypes on the expression levels of core circadian clock genes. Acknowledging this concern, we have revised the Results section to include a clarification. Specifically, we state that the differences in overall transcriptional activity between breast cancer subtypes may impact the expression of clock genes, independent of their circadian functionality (see page 9, lines 276–281). This addition helps to contextualize our findings and address the variability that might influence gene expression patterns observed in our study.

11. It seems that growth rate might have the potential to influence the bioluminescence reporter recordings, even without invoking the coupling between cells, very fast cell cycling like in the MDAMB231 cells might make the clock appear "worse" than single cell recordings would suggest. This might also influence the drug sensitivity (also in comparison to Lee et al). We don't suggest replicating the findings with single cell recording. However, this should be discussed explicitly.

We thank the reviewer for highlighting the potential confounding effect of varying growth rates on our bioluminescence reporter recordings and drug sensitivity. In the revised Discussion section, we have now explicitly addressed this issue and emphasized the necessity of more advanced methods such as single-cell imaging techniques to obtain precise measurements of circadian dynamics and drug responses (pages 15 and 16, lines 476–489).

Minor comments

1. The authors write: "Through the introduction of the "chronosensitivity index", we provide a robust metric for quantifying the circadian clock's impact on differentiating between high and low drug sensitivity groups". The ratio of between to within cluster distances is one of the many standard metrics to assess the quality of clustering, if not being the most advanced. Why they wouldn't just call it that?

Thank you for your valuable feedback regarding the use of the term “chronosensitivity index” in our study. We acknowledge that the ratio of between to within cluster distances is a well-established metric for evaluating clustering quality, and we apologize for any confusion our terminology might have caused. Our intention in introducing “chronosensitivity index” was to provide a term that intuitively connects to circadian variations in drug sensitivity, making it particularly relevant to our research focus.

In the revised manuscript, we have now made it clearer that this term builds upon the established clustering metric, with specific adaptations that reflect the circadian aspects of our study (page 12, lines 356–360). We have also detailed how this metric is suited to chronopharmacology.

We understand the importance of using precise and recognized terminology and are willing to further adapt the name and definition of the "chronosensitivity index" to align with standard definitions, should the reviewer and editor recommend this approach.

2. Line 671: They did not mean to cite Balsalobre's paper here.

We thank the reviewer for bringing this to our attention. We meant to cite Barretina et al.⁸ and have corrected it in the revised manuscript (page 25, now lines 755, 756).

****Reviewer #3****

The manuscript by Ector and colleagues, titled "Circadian Clock Features Define Novel Breast Cancer Types and Shape Drug Sensitivity," focuses on the application of circadian oscillators' properties to redefine subtypes of breast cancer, as suggested by the title. The authors set a highly ambitious goal to redefine classical breast cancer phenotypes by combining in-depth phenotyping of cellular circadian oscillators with analyses of cell growth, genomic properties, and drug responses. To achieve this, the authors applied their own recently published novel methodological approach (from this year's Nature Communications publication by the same group), which they extend in terms of analysis, to 19 immortalized cell lines (both malignant and benign). Multiresolution analysis was

employed to assess different time components, followed by an assessment of the signal-to-noise ratio for the circadian component. This parameter was further validated using the U2OS cell line, which lacks either CRY proteins alone or both Cry1/Cry2. These state-of-the-art analyses, designed to define circadian clock-based cancer cell phenotypes, were completed by assessing phase coherence between two anti-phasic circadian reporters (Bmal1-luc and Per2-luc) introduced into each cell line, to further define the coupling strength for each cancer subtype. These analyses were then combined with mutational assessments.

The proposed clock phenotyping was further correlated with cell growth rate. Strikingly, the circadian phenotyping proposed here did not fully overlap with classical subtyping, opening new possibilities for precision medicine based on optimal therapeutic sensitivity as defined by circadian phenotyping. To further explore this novel classification, the authors screened a wide spectrum of classical anticancer agents to correlate drug responsiveness with clock parameters. The correlations varied widely depending on the nature of the compounds, with distinct contributions from each of the core clock genes, which were analyzed individually, identifying Rora and Cry2 as the most significant contributors.

The circadian phenotyping proposed here holds outstanding promise, not only for understanding the roles of core clock machinery in breast cancer progression but also for developing personalized therapeutic approaches (including chronotherapy) better tailored to the malignancy characteristics. In summary, this study represents a valuable advancement from both basic and preclinical perspectives; it is well-designed and clearly presented.

We sincerely thank the reviewer for their insightful feedback and comments on our study. We have carefully addressed the concerns by expanding on our Discussion and potential next steps. We further made all code and datasets associated to this study publicly available to ensure its future use in other studies.

Minor comments

1. While immortalized cells are an indispensable model for basic studies and enable high-throughput analyses like those conducted by the authors, using primary cells may represent an important next step towards the preclinical stage. Although these approaches are beyond the scope of the current study, future application of the proposed methodology to primary human cells derived from malignant and non-malignant tissue biopsies, and the feasibility of such an approach, could be discussed.

We strongly agree with the reviewer that primary cells from malignant and non-malignant tissue biopsies represent an intriguing and important next step to bridge to

preclinical studies. Although this approach was beyond the scope of the current study, we now discuss future applications of our proposed methodology to primary cells in our revised Discussion section on page 16 in lines 490–496.

2. A recent publication by Manella and colleagues (PMID: 34625543) presents a novel method for rapidly establishing clock-resetting properties (CIRCA-Scope). Do the authors consider this method as complementary to the analyses presented here (e.g., clock resetting upon therapeutic administration in a time-of-day dependent manner)? If so, this point could be discussed.

We agree with the reviewer that CIRCA-Scope offers valuable insights into clock-resetting properties in a time-of-day dependent manner. We have now included a new section in the main text where we discuss the potential integration of time-of-day dependent treatments with CIRCA-Scope for future research directions on page 16 in lines 486–489. Together, this has the potential to not only validate drug candidates that likely act in a circadian-dependent manner, but also to evaluate their effects on the circadian clock itself.

3. One of the major novelties of this work is the method for in-depth circadian phenotyping combined with additional parameters. It would be of great value to the field if the authors made the developed methodology accessible to the reader in a straightforward manner, upon publication of the manuscript.

Thanks for this great suggestion. We have now uploaded the developed methodology for in-depth circadian phenotyping and all other computational code to the public repository Zenodo under the identifier <https://doi.org/10.5281/zenodo.14657750>.

****Reviewer-independent changes to the manuscript:****

1. We identified an error in the calculation of the cell lines' growth rates, which were initially derived from 2 days of the live-imaging recordings instead of the intended 4 days. This has been corrected in the revised manuscript, and all relevant figures have been updated (Fig 3A, B; Fig 4C–F, H, I; Fig EV2A, B; Fig EV3; Fig EV4; and Appendix Fig S2). Despite this adjustment, the results involving growth rate parameters remain unchanged due to the relatively minor differences between the growth rates in the initial and revised calculations.
2. We have added numbering to the equations in the Methods section of the revised manuscript to facilitate easier referencing.

3. We expanded the explanation of the parameters in equations 4 and 5 in the Methods section (page 22, lines 665–668) for enhanced clarity.
4. A plotting error in Fig 3F, which caused incorrect y-axis values, has been corrected. This adjustment does not alter the overall message.

References

1. Qu, L. & Pei, Y. A Comprehensive Review on Discriminant Analysis for Addressing Challenges of Class-Level Limitations, Small Sample Size, and Robustness. *Processes* **12**, 1382 (2024).
2. Sharma, A. & Paliwal, K.K. Linear discriminant analysis for the small sample size problem: an overview. *International Journal of Machine Learning and Cybernetics* **6**, 443-454 (2015).
3. Lee, Y. *et al.* Time-of-day specificity of anticancer drugs may be mediated by circadian regulation of the cell cycle. *Science Advances* **7**, eabd2645 (2021).
4. Talamanca, L., Gobet, C. & Naef, F. Sex-dimorphic and age-dependent organization of 24-hour gene expression rhythms in humans. *Science* **379**, 478-483 (2023).
5. Zhang, R., Lahens, N.F., Ballance, H.I., Hughes, M.E. & Hogenesch, J.B. A circadian gene expression atlas in mammals: implications for biology and medicine. *Proc Natl Acad Sci U S A* **111**, 16219-16224 (2014).
6. Mure, L.S. *et al.* Diurnal transcriptome atlas of a primate across major neural and peripheral tissues. *Science* **359** (2018).
7. Liu, X.-L., Duan, Z., Yu, M. & Liu, X. Epigenetic control of circadian clocks by environmental signals. *Trends in Cell Biology* **34**, 992-1006 (2024).
8. Barretina, J. *et al.* The Cancer Cell Line Encyclopedia enables predictive modelling of anticancer drug sensitivity. *Nature* **483**, 603-607 (2012).

10th Feb 2025

Manuscript Number: MSB-2024-12581R

Title: Circadian clock features define novel subtypes among breast cancer cells and shape drug sensitivity

Author: Carolin Ector

Jeff Didier

Sébastien De Landtsheer

Malthe Nordentoft

Christoph Schmal

Ulrich Keilholz

Hanspeter Herzog

Achim Kramer

Thomas Sauter

Adrián Granada

Dear Dr Granada,

Thank you for sending us your revised manuscript. We have now heard back from the two reviewers who agreed to evaluate your revised study. As you will see below, the reviewers are satisfied with the performed revisions and support publication. Before we can formally accept the manuscript for publication, we would ask you to address some remaining editorial-level issues listed below.

1. The funding information for the Berlin School of Integrative Oncology (BSIO) is missing from the online submission system and needs to be added.
2. "Disclosure statement and competing interests" needs to be renamed to "Disclosure and Competing Interests Statement".
3. Author contribution section needs to be removed from the manuscript file.
4. "Glossary" needs to be removed from the manuscript file.
5. In the Author checklist, regarding the question, 'For every figure, are statistical tests justified as appropriate? Do the data meet the assumptions of the tests (e.g., normal distribution)? Describe any methods used to assess it. Is there an estimate of variation within each group of data? Is the variance similar between the groups that are being statistically compared?', please specify in which section this information is available (in the pink column).
6. EV tables: As these two tables are quite complex, they need to be updated to EV datasets. Source file names, titles, legends and manuscript callouts all need to be updated to Dataset EV1-EV2.
7. Section order should be corrected: Title page - Abstract & Keywords - Introduction - Results - Discussion - Methods - Data Availability - Acknowledgements - Disclosure and Competing Interests Statement - References - Figure Legends - Table(s) - Expanded View Figure Legends.
8. Source data: Source data files need to be saved in a scheme one figure/folder and then uploaded as .zip files. E.g. all the Source data files for Figure 1 need to be saved in a single folder and this needs to be zipped and then uploaded as "SD figure 1.zip" file.
9. Please address the following comments related to figure legends:
 - Please note that the box plots need to be defined in terms of minima, maxima, centre, bounds of box and whiskers, and percentile in the legends of figures EV3B.
 - Please note that information related to n is missing in the legend of figure 2H.
 - Please note that the error bars are not defined in the legend of figure 2H.

When you resubmit your manuscript, please download our CHECKLIST (<https://bit.ly/EMBOPressAuthorChecklist>) and include the completed form in your submission. *Please note* that the Author Checklist will be published alongside the paper as part of the transparent process (<https://www.embopress.org/page/journal/17444292/authorguide#transparentprocess>)

Click on the link below to submit your revised paper.

Thank you for submitting this interesting paper to Molecular Systems Biology.

Kind regards,
Jingyi

Jingyi Hou, PhD
Senior Editor
Molecular Systems Biology

If you do choose to resubmit, please click on the link below to submit the revision online before 12th Mar 2025.

IMPORTANT: When you send your revision, we will require the following items:

1. the manuscript text in LaTeX, RTF or MS Word format
2. a letter with a detailed description of the changes made in response to the referees. Please specify clearly the exact places in the text (pages and paragraphs) where each change has been made in response to each specific comment given
3. three to four 'bullet points' highlighting the main findings of your study
4. a short 'blurb' text summarizing in two sentences the study (max. 250 characters)
5. a 'thumbnail image' (550px width and max 400px height, Illustrator, PowerPoint or jpeg format), which can be used as 'visual title' for the synopsis section of your paper.
6. Please include an author contributions statement after the Acknowledgements section (see <https://www.embopress.org/page/journal/17444292/authorguide#manuscriptpreparation>)
7. Please complete the CHECKLIST available at (<https://bit.ly/EMBOPressAuthorChecklist>). Please note that the Author Checklist will be published alongside the paper as part of the transparent process (<https://www.embopress.org/page/journal/17444292/authorguide#transparentprocess>).
8. When assembling figures, please refer to our figure preparation guideline in order to ensure proper formatting and readability in print as well as on screen:
<https://bit.ly/EMBOPressFigurePreparationGuideline>
See also figure legend guidelines: <https://www.embopress.org/page/journal/17444292/authorguide#figureformat>
9. Please note that corresponding authors are required to supply an ORCID ID for their name upon submission of a revised manuscript (EMBO Press signed a joint statement to encourage ORCID adoption). (<https://www.embopress.org/page/journal/17444292/authorguide#editorialprocess>)
Currently, our records indicate that the ORCID for your account is 0000-0003-0537-9091.

Please click the link below to modify this ORCID:
Link Not Available

10. Include a Reagents and Tools Table as part of the Methods section, which can be downloaded from our author guidelines (<https://www.embopress.org/page/journal/17444292/authorguide#structuredmethods>)

*** PLEASE NOTE *** As part of the EMBO Press transparent editorial process initiative (see our Editorial at <https://dx.doi.org/10.1038/msb.2010.72>), Molecular Systems Biology will publish online a Review Process File to accompany

accepted manuscripts. When preparing your letter of response, please be aware that in the event of acceptance, your cover letter/point-by-point document will be included as part of this File, which will be available to the scientific community. More information about this initiative is available in our Instructions to Authors. If you have any questions about this initiative, please contact the editorial office (msb@embo.org).

Reviewer #2:

The authors have addressed our comments satisfactorily. I am looking forward to read the paper.

Reviewer #3:

The authors fully and satisfactory responded to all the points of criticism.

Editorial-level issues

1. The funding information for the Berlin School of Integrative Oncology (BSIO) is missing from the online submission system and needs to be added.
C.E. received a stipend from the BSIO graduate school, which was not linked to a specific grant number. After consulting with BSIO, it was agreed that their support would be acknowledged solely through the affiliations rather than in the acknowledgments.
2. "Disclosure statement and competing interests" needs to be renamed to "Disclosure and Competing Interests Statement" .
We renamed the section to “Disclosure and Competing Interests Statement”
3. Author contribution section needs to be removed from the manuscript file.
We removed the “Author contribution” section from the manuscript file.
4. "Glossary" needs to be removed from the manuscript file.
We removed the “Glossary” section from the manuscript file.
5. In the Author checklist, regarding the question, ' For every figure, are statistical tests justified as appropriate? Do the data meet the assumptions of the tests (e.g., normal distribution)? Describe any methods used to assess it. Is there an estimate of variation within each group of data? Is the variance similar between the groups that are being statistically compared?' , please specify in which section this information is available(in the pink column).
We have now specified in which section this information is available in the Author Checklist.
6. EV tables: As these two tables are quite complex, they need to be updated to EV datasets. Source file names, titles, legends and manuscript callouts all need to be updated to Dataset EV1-EV2.
We renamed Table EV1-EV2 to Dataset EV1-EV2 in the file names, figure legends and throughout the manuscript.
7. Section order should be corrected: Title page - Abstract & Keywords - Introduction - Results - Discussion - Methods - Data Availability - Acknowledgements - Disclosure and Competing Interests Statement - References - Figure Legends - Table(s) - Expanded View Figure Legends.
We adapted the section order accordingly. Please note, we included the legend to “Box 1” within the ‘Figure Legends’ sections.

8. Source data: Source data files need to be saved in a scheme one figure/folder and then uploaded as .zip files. E.g. all the Source data files for Figure 1 need to be saved in a single folder and this needs to be zipped and then uploaded as "SD figure 1.zip" file.

We now uploaded the source data files zipped individually for each figure.

9. Please address the following comments related to figure legends:

- Please note that the box plots need to be defined in terms of minima, maxima, centre, bounds of box and whiskers, and percentile in the legends of figures EV3B.

We now included the information to the figure legend of EV3B. In the legend of EV3C, we refer to the legend of EV3B regarding the box information to avoid redundancy.

- Please note that information related to n is missing in the legend of figure 2H.

We now included the number of samples n in the figure legend of 2H.

- Please note that the error bars are not defined in the legend of figure 2H.

We now improved and expanded on our explanation of the error bars in the figure legend of 2H.

Reviewer #2

The authors have addressed our comments satisfactorily. I am looking forward to read the paper.

Reviewer #3

The authors fully and satisfactory responded to all the points of criticism.

We sincerely appreciate the reviewers' efforts and valuable feedback throughout the review process of our manuscript. We are pleased that they are satisfied with our revisions and have no further concerns.

12th Feb 2025

Manuscript number: MSB-2024-12581RR

Title: Circadian clock features define novel subtypes among breast cancer cells and shape drug sensitivity

Dear Dr Granada,

Thank you again for sending us your revised manuscript. We are now satisfied with the modifications made and I am pleased to inform you that your paper has been accepted for publication.

Yours sincerely,
Jingyi

Jingyi Hou, PhD
Senior Editor
Molecular Systems Biology
